# SAMHD1 depletion restricts SARS-CoV-2 infection by suppressing HNF1-dependent ACE2 expression in lung epithelial cells

Pak-Hin Hinson Cheung[1]*, Pearl Chan[1], Hua Yang[1], Krisztina Ambrus[2], Shravya Honne[2], Baek Kim[2], Stanley Perlman[1], Li Wu [1]*

1 Department of Microbiology and Immunology, Carver College of Medicine, University of Iowa, Iowa City, Iowa, United States of America, 2 Department of Pediatrics, School of Medicine, Emory University, Atlanta, Georgia, United States of America

* pakhinhinson-cheung@uiowa.edu (PHC); li-wu@uiowa.edu (LW)

## Abstract

Sterile alpha motif and histidine-aspartate domain-containing protein 1 (SAMHD1) restricts a broad spectrum of viruses through multifaceted mechanisms. It also limits spontaneous- and virus-induced innate immune responses by suppressing proinflammatory cytokine and type-I interferon (IFN-I) production. Some viruses escape SAMHD1 restriction and utilize SAMHD1-mediated innate immune suppression to establish effective infection through IFN antagonism. Our previous studies showed that SAMHD1 is a proviral factor facilitating replication of severe acute respiratory syndrome coronavirus-2 (SARS-CoV-2) in human macrophages, monocytic THP-1 and epithelial-like HEK293T cell lines by suppressing IFN responses. However, it is unclear about the function of SAMHD1 in lung epithelial cells during SARS-CoV-2 infection. Here, we report that SAMHD1 knockout (KO) restricts SARS-CoV-2 replication in lung epithelial Calu-3 cells by suppressing endogenous expression of the viral receptor angiotensin-converting enzyme 2 (ACE2) via hepatocyte nuclear factor 1-alpha (HNF1α) and HNF1β. Using pseudotyped SARS-CoV-2 and lentiviral vectors, we found that SARS-CoV-2 spike protein-mediated viral entry was suppressed in SAMHD1 KO Calu-3 cells. SAMHD1 KO repressed ACE2 expression in Calu-3 cells at mRNA and protein levels. Functional analyses revealed that HNF1α and HNF1β were crucial for the endogenous ACE2 expression in Calu-3 cells. Additionally, SAMHD1 KO led to a reduction in the expression levels and ACE2-promoting function of HNF1α and HNF1β. Inhibition of IFN antiviral response by baricitinib, a Janus kinase 1 and 2 (JAK 1/2) inhibitor, did not revert the suppression of SARS-CoV-2 in SAMHD1 KO Calu-3 cells. SAMHD1 knock-in and deoxynucleoside supplementation experiments indicated that SAMHD1 expression and dNTP pool balance collectively regulated HNF1-mediated ACE2 expression in Calu-3 cells. Our findings demonstrate that SAMHD1 depletion hinders HNF1-mediated ACE2 expression and SARS-CoV-2 replication in Calu-3 cells via a novel mechanism beyond its IFN-suppressive function.

**Data availability statement:** All relevant data are within the manuscript and its Supporting Information files.

**Funding:** This work was supported in part by National Institutes of Health grants R21AI181742 (to L.W.), R01AI129269 (to S.P.), and R01AI136581 and R01AI162633 (to B.K.). L.W. is also supported by the NIH grants R01AI189220, R33AI169659, and P30CA086862-25S1. The funders had no role in study design, data collection and analysis, decision to publish, or preparation of the manuscript.

**Competing interests:** The authors have declared that no competing interests exist.

## Author summary

During viral infection, SAMHD1 acts as a viral restriction factor and a suppressor of the innate immune system, controlling viral replication while also ensuring immune homeostasis. The innate immune suppressive function of SAMHD1 can be proviral for some viruses. SAMHD1 KO has been shown to hinder SARS-CoV-2 infection in human macrophages, THP-1 and HEK293T cell lines by promoting IFN response, but its role in lung epithelial cells is unclear. Here, we demonstrated that SAMHD1 KO impeded SARS-CoV-2 replication in human lung epithelial Calu-3 cells by downregulating the expression of the major viral receptor ACE2. We found that SAMHD1 KO suppressed HNF1-medaited ACE2 expression that was required for spike protein-mediated SARS-CoV-2 entry. However, inhibiting IFN signaling in SAMHD1 KO Calu-3 cells was not sufficient to revert SARS-CoV-2 replication. Restoring SAMHD1 expression and increasing dNTP intracellular pool in Calu-3 cells suggested multifaceted mechanisms contributing to the regulation of HNF1-mediated ACE2 expression. Our findings shed light on the differential proviral function of SAMHD1 in ACE2 expressing cells and suggest that SAMHD1 expression can facilitate SARS-CoV-2 infection beyond enhancing IFN antagonism.

## Introduction

SAMHD1 is the only deoxynucleoside triphosphohydrolase (dNTPase) of mammalian cells to be a restriction factor against human immunodeficient virus type 1 (HIV-1) infection in resting CD4[+] T cells and terminally differentiated immune cells, such as macrophages and dendritic cells [1–3]. Increasing evidence suggests that SAMHD1 restricts different viruses including human T cell leukemia virus type 1 [4], DNA viruses such as herpesviruses (herpes simplex virus 1, human cytomegalovirus, and Epstein-Barr virus), hepatitis B virus, human papillomavirus and poxviruses [5–10] as well as RNA viruses including influenza A virus, enteroviruses, and flaviviruses [11–13]. The viral restriction functions of SAMHD1 encompass both dNTPase-dependent and independent mechanisms according to the target virus and the contexts of the host cells such as cell type, differentiation status, IFN-I response and the expression of other host restriction factors that cooperate with SAMHD1 in cells [14].

SAMHD1 suppresses innate immune responses [15]. Loss-of-function mutation of SAMHD1 correlates with the development of the Aicardi-Goutières syndrome which is an autoimmune disease implicating central nervous system [16]. SAMHD1 prevents spontaneous activation of innate immune responses to self-RNA and self-DNA species that respectively activate retinoic acid-inducible gene I (RIG-I)-like receptor signaling and cyclic GMP-AMP synthase (cGAS)-stimulator of interferon genes (STING) signaling [17–24]. Similarly, SAMHD1 prevents inflammation- or virus-induced innate immune responses [25,26]. SAMHD1 suppresses pattern recognition receptor signal pathways by interacting with various signal proteins such as mitochondrial



antiviral-signaling protein (MAVS), inhibitor of nuclear factor kappa-B kinase subunit alpha (IKKα), IKKβ, IKKε as well as transcription factors including p100 and p105 in the NF-κB pathway which drives proinflammatory cytokine expression and interferon regulatory factor 7 (IRF7) which promotes IFN-I expression [25,27,28].

SARS-CoV-2 caused the COVID-19 pandemic, claiming more than 7 million lives worldwide [29,30]. SARS-CoV-2 suppresses IFN antiviral responses through multiple strategies to facilitate viral infection [31]; however, the precise mechanisms remain to be elucidated. Previously, we reported that SARS-CoV-2 infection was suppressed upon depletion of SAMHD1 expression in HEK293T, differentiated THP-1 cells, and primary human macrophages [32]. Suppression of IFN signaling of SAMHD1-defective HEK293T cells by baricitinib, a JAK1/2 inhibitor, alleviated the repression of SARS-CoV-2 replication, suggesting that SARS-CoV-2 exploits SAMHD1 for IFN antagonism, thereby facilitating its infection [32]. It remains to be determined if SAMHD1-mediated immunosuppression promotes SARS-CoV-2 replication in the primary target cell type, such as lung epithelial cells expressing the major viral entry factors ACE-2 and transmembrane protease, serine 2 (TMPRSS2) [33,34].

In this study, we found that SAMHD1 KO in Calu-3 cells conferred resistance to SARS-CoV-2 infection, independent of IFN signaling, by reducing the expression of transcription factors HNF1α and HNF1β, which in turn lowered ACE2 levels and inhibited S-mediated viral entry. Our findings shed light on a novel function of SAMHD1 in regulating ACE2 expression and SARS-CoV-2 infection in human lung epithelial cells.

## Results

### Generation and characterization of stable SAMHD1 KO Calu-3 cell lines

To generate SAMHD1 KO Calu-3 cells, clustered regularly interspaced short palindromic repeats (CRISPR) and CRISPR-associated protein 9 (Cas9) technology based on the lentiCRISPRv2 system was employed [35]. Two target sequences of the single guide RNAs (sgRNAs) were obtained from CHOPCHOP v3 with no off-target effect at less than four mismatches [36]. With sgRNA 1 and sgRNA 2, the *Streptococcus pyogenes* Cas9 nuclease (SpCas9) was targeted to the SAMHD1 gene, resulting in cleavage sites that encoded [11]Lysine and [150]Leucine, respectively (Fig 1A). [11]Lysine was the first residue of the nuclear localizing signal (NLS) of SAMHD1 protein and [150]Leucine was in the HD domain before the dNTPase active site. For empty vector control, lentiCRISPR v2-puro empty vector which did not express sgRNA was used.

Calu-3 cells were transduced with the lentiviral vectors, selected by puromycin and clonal purified with limiting dilution. Several purified cell clones were obtained and tested for SAMHD1 protein expression (Fig 1B). We obtained clone 4 and clone 3 of the respective cell populations treated with sgRNA1 or sgRNA2 having the lowest expression of SAMHD1 compared with the Ctrl clones (Fig 1B). Therefore, clone 1 of the Ctrl population, clone 4 of the sgRNA1-treated population and clone 3 of the sgRNA2-treated population were used for the subsequent study and named Ctrl, KO1 and KO2, respectively.

The genomic DNA sequences at the SpCas9 cleavage sites of KO1 cells (S1A Fig) and KO2 cells (S1B Fig) were obtained through Sanger sequencing and compared with Ctrl cells. We found that a single nucleotide insertion occurred near the SpCas9 cleavage sites, which frameshifted the SAMHD1 coding sequence thereafter (Red arrows, S1A and S1B Fig). It was noted that multiple signal peaks were observed after the SpCas9 cleavage sites of KO1 and KO2 which may indicate the existence of more than one mutant in the clonal purified population.

To confirm whether SAMHD1 KO functionally affected intracellular dNTP levels, cellular dATP, dTTP, dGTP and dCTP levels were measured (S1C Fig) [37]. We found that the intracellular levels of all four dNTP of KO1 and KO2 were significantly higher than those in the Ctrl cells (S1C Fig). This result suggested that KO1 and KO2 were deprived of dNTPase activity of SAMHD1.

SAMHD1 depletion may enhance innate immune response even in the absence of extrinsic stimuli depending on cell types [19,24,25,32,38–40]. To examine if SAMHD1 KO promoted innate immune responses of Calu-3 cells, the

**A**

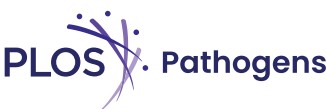

**B**

**C**

**Fig 1. Generation and characterization of SAMHD1 KO Calu-3 cell lines. (A)** Schematic diagram of human SAMHD1 protein domains showing the target sites of sgRNA1 or sgRNA2. **(B)** SAMHD1 expression of Calu-3 cells transduced to express SpCas9 without the expression of any sgRNA (Ctrl, two single clones) or with the expression of sgRNA1 (five single clones) or sgRNA2 (three single clones). GAPDH detection was used as normalization control. The relative band intensities of SAMHD1 were calculated by dividing them with GAPDH bands and were normalized to Ctrl clone 1. Clone 1 of Ctrl, clone 4 of sgRNA1- and clone 3 of sgRNA 2- treated groups were chosen for all subsequent experiments and respectively named Ctrl, KO1 and KO2. **(C)** Levels of *IL-6, IFNβ, OAS1* and *ISG15* mRNA were measured with RT-qPCR. 18S RNA was used as internal control. Biological sextuplicate experiments were performed. One-way ANOVA multiple comparisons test was used to evaluate the statistical significance of the difference between Ctrl and KO1 or KO2. * $P < 0.05$, *** $P < 0.001$, **** $P < 0.0001$. ns, not significant.



mRNA levels of proinflammatory cytokine *interleukin-6* (*IL-6*) as well as *IFNβ* and two IFN-stimulated genes (ISGs), *2'-5'-oligoadenylate synthetase 1* (*OAS1)* and *ISG15,* of KO1 and KO2 were detected by RT-qPCR and compared with Ctrl cells (Fig 1C). We found that KO1 and KO2 cells constitutively expressed higher levels of IL-6 in the absence of any extrinsic stimuli. Interestingly, we found that the expression levels of *IFNβ*, *OAS1* and *ISG15* were significantly upregulated in KO2, but not in KO1 clone (Fig 1C). Thus, SAMHD1 KO promoted proinflammatory cytokine production from Calu-3 cells in both KO1 and KO2 clones, while strong constitutive IFN-I response was only observed in KO2 clone.

## SAMHD1 KO restricts SARS-CoV-2 replication independent of IFN signaling

To test if SAMHD1 KO affected SARS-CoV-2 replication in Calu-3 cells, kinetics of ancestral SARS-CoV-2 (Wuhan-Hu-1) replication was examined in Ctrl, KO1 and KO2 cells. We found that the kinetics of infectious virus production were suppressed in both KO1 and KO2 (Fig 2A) despite only KO2 cells having a higher constitutive IFN-I response than Ctrl cells (Fig 1C). Similarly, significant suppression of the levels of released viral genome was observed in both KO1 and KO2 clones (Fig 2B). The expressions of intracellular viral nucleoprotein (N) and spike protein (S) were significantly reduced in the infected KO1 and KO2 cells (Fig 2C). Moreover, intracellular viral RNA levels were suppressed in the infected KO1 and KO2 cells which did not produce higher levels of IFNβ, IFNλ or OAS1 in response to SARS-CoV-2 infection (Fig 2D). To confirm if IFN signaling contributed to the hindered replication of SARS-CoV-2 in SAMHD1 KO Calu-3 cells, infected cells were treated with or without baricitinib to test if suppressing IFN signaling can rescue SARS-CoV-2 replication in KO1 and KO2 cells. We found that baricitinib did not rescue SARS-CoV-2 replication in Calu-3 cells in terms of released infectious viral titer (Fig 2E), intracellular viral N protein (Figs 2F, S2A and S2B), and intracellular viral *N* mRNA (Fig 2G), although baricitinib efficiently suppressed virus-induced *OAS1* expression in Ctrl and KO2 cells (Fig 2G) as well as virus-induced Ser727-phosphorylated signal transducer and activator of transcription 1 (STAT1) (Figs 2F, S2A and S2B). Therefore, SAMHD1 KO suppressed SARS-CoV-2 replication in Calu-3 cells independent of IFN signaling.

## SAMHD1 KO suppresses spike protein-mediated viral entry of SARS-CoV-2

To understand how SAMHD1 KO suppressed SARS-CoV-2 infection, we employed a previously developed single cycle SARS-CoV-2 which was S-defective and pseudotyped with vesicular stomatitis virus glycoprotein (VSV-G), namely, ΔS-VRP(G) [41]. ΔS-VRP(G) infection bypasses S-protein mediated viral entry and infects cell types that do not support authentic SARS-CoV-2 infection. Firstly, we confirmed that remdesivir, a nucleoside analogue inhibiting viral RNA-dependent RNA polymerase (RdRP) [42], can efficiently suppress intracellular viral RNA levels (N and E genes) of ΔS-VRP(G) in Calu-3 Ctrl cells, suggesting that ΔS-VRP(G) infection depended on RdRP activity (Fig 3A). Next, we asked if SAMHD1 deficiency suppressed ΔS-VRP(G) infection similarly as that observed in the authentic SARS-CoV-2 infections (Fig 2). Interestingly, we found that the intracellular viral *N* and *E* RNA levels of ΔS-VRP(G) were not suppressed in KO1 or KO2 cells compared with Ctrl cells at 24 and 48 hr post-infection (hpi) (Fig 3B). Similarly, we observed that intracellular viral N protein expression by ΔS-VRP(G) infection was minimally affected in KO1 and KO2 cells (Figs 3C, S3A and S3B). ΔS-VRP(G) expresses a dual reporter gene (*Gaussia* luciferase-T2A-neon GFP) through the transcriptional regulatory sequence of S gene [41]. Detection to the *Gaussia* luciferase activity reflects the infectivity of ΔS-VRP(G) [41]. We found that ΔS-VRP(G) infection produced similar levels of *Gaussia* luciferase in Ctrl, KO1 and KO2 cells (Fig 3D). Although ΔS-VRP(G) infection had a trend of producing less *Gaussia* luciferase in KO2 cells than KO1 and Ctrl, we did not observe significant reductions in both KO1 and KO2 cells (Fig 3D). Therefore, unlike authentic SARS-CoV-2, SAMHD1 deficiency did not inhibit ΔS-VRP(G) infection.

One possible explanation for the selective suppression of authentic SARS-CoV-2 infection but not ΔS-VRP(G) in SAMHD1 KO cells was due to abolished S-protein-mediated viral entry, which was not required for ΔS-VRP(G) infection. To validate the possibility, we employed a SARS-CoV-2 S protein-pseudotyped HIV-1-based reporter virus, namely HIV-1-spike-Luc/ZsGreen [43,44]. Successful infection of HIV-1-spike-Luc/ZsGreen depended on S-protein mediated viral

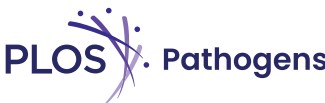

**Fig 2. Suppression of SARS-CoV-2 infection by SAMHD1 depletion independent to IFN antiviral response. (A, B)** Calu-3 Ctrl, KO1 or KO2 cells were infected with authentic SARS-CoV-2 at MOI = 0.05. Culture supernatant was collected at 0, 24, 48 or 72 hpi. **(A)** Infectious titer or **(B)** viral RNA copy number of the conditioned media were respectively measured by plaque assay or RT-qPCR. Statistical comparison was performed at each time point between Ctrl and KO1 or KO2. **(C)** Viral N and S proteins of infected Ctrl, KO1 or KO2 cells (SARS-CoV-2, MOI = 1) were detected at 24 hpi.

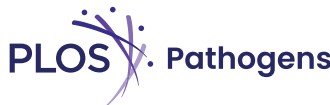

GAPDH was detected as an input control. The relative band intensities of viral N and viral S were calculated by dividing them with GAPDH bands and were normalized to Ctrl cells. **(D)** The levels of viral RNA (*N*), *IFNβ* mRNA, *IFNλ* mRNA and *OAS1* mRNA of infected Ctrl, KO1 or KO2 cells (SARS-CoV-2, MOI = 1, 0 or 24 hpi) were quantified with RT-qPCR. 18S RNA was used as internal control. **(E, F** and **G)** Calu-3 Ctrl, KO1 or KO2 cells treated with 10 μM baricitinib or DMSO (added 4 hr before infection) were infected with authentic SARS-CoV-2 at MOI = 1. At 24 hpi, culture supernatant, cellular protein and cellular RNA were harvested and respectively tested with **(E)** plaque assay, **(F)** Western blot or **(G)** RT-qPCR. For (F), viral N protein, total STAT1 or S727-phosphorylated STAT1 (p-STAT1) were detected and GAPDH was used as input control. The relative band intensities of viral N, STAT1, or p-STAT1 were calculated by dividing them with GAPDH bands and were normalized to Ctrl cells treated with DMSO. For (G), viral *N* RNA and *OAS1* mRNA were detected. 18S RNA was used as normalization control. For (A), (B), (D) and (G), biological triplicate experiment was performed. For (E), biological quadruplicate experiment was performed. For (A), (B), (D), (E) and (G), two-way ANOVA multiple comparisons test was performed. * $P < 0.05$, ** $P < 0.01$, *** $P < 0.001$, **** $P < 0.0001$. ns, not significant.

entry and the subsequent viral gene expression driven by HIV-1 single-cycle infection. The infectivity of HIV-1-spike-Luc/ZsGreen can be measured by detecting the expression of the firefly luciferase 2 [43,44]. We found that HIV-1-spike-Luc/ZsGreen infected KO1 or KO2 cells produced significantly less luciferase activity than the Ctrl cells (Fig 3E). This suggested that SAMHD1 KO suppressed S protein-mediated viral entry, which in turn inhibited infection of authentic SARS-CoV-2 and HIV-1-spike-Luc/ZsGreen but not ΔS-VRP(G).

### SAMHD1 KO inhibits the expression of viral receptor ACE2 of Calu-3 cells

To gain mechanistic insight into how SAMHD1 KO suppressed spike-protein mediated viral entry, the transcriptomic profiles of Ctrl, KO1 and KO2 cells were obtained by mRNA-sequencing (mRNA-seq). Firstly, principal component analysis (PCA) was performed to cluster Ctrl, KO1 and KO2 cells. Biological triplicate experiment was performed. Nine PCs were therefore obtained, in which PC1 (61%) and PC2 (37%) covered 98% of the total variance (S4A Fig). Ctrl, KO1 and KO2 cells were well clustered based on PC1 and PC2 (S4B Fig). Along PC1 axis, KO1 and KO2 cells were clearly separated from Ctrl cells in the same direction. KO2 cells were separated from KO1 and Ctrl cells along the PC2 suggesting clonal specific transcriptomic profile for KO2.

A PCA biplot on the top ten most influential loading genes for the PCA clustering was plotted (S4C Fig). The PCA biplot indicated that the major loading contributors of PC1 were cell surface proteins including carcinoembryonic antigen-related cell adhesion molecule 6 (CEACAM6), Galectin 4 (LGALS4), and gap junction alpha-1 (GJA1), and proteins implicating extracellular matrix including cystatin E/M (CST6) and family with sequence similarity 20 member C (FAM20C). For PC2, the expression changes of ISGs including OAS2, interferon alpha inducible protein 27 (IFI27), interferon-induced transmembrane protein 1 (IFITM1) and IFI6 as well as major histocompatibility complex class II DP beta 1 (HLA-DPB1) distinguished KO2 from Ctrl and KO1 but had minimal contribution to PC1 axis (S4C Fig). In summary, KO1 and KO2 were distinguished from Ctrl along PC1 which was mainly influenced by the mRNA expression of genes implicating cell surface and extracellular matrix while KO2 was further distinguished from Ctrl and KO1 along PC2 which was mainly influenced by the expressions of ISGs.

Next, by analyzing the differential expressed genes (DEGs) implicated in both KO1 and KO2 having more than 2-fold change and with false discovery rate (FDR)-adjusted P values < 0.01 (S1 Table), we found that 806 DEGs were commonly observed in KO1 and KO2 compared with Ctrl cells (Fig 4A). Among the 806 DEGs, 681 DEGs were of the same direction of change (S2 Table). We searched for known and validated entry factors of SARS-CoV-2 infection. We found that the expressions of ACE2 and TMPRSS2 were decreased in both KO1 and KO2 (Fig 4B). Both ACE2 and TMPRSS2 transcripts were more suppressed in KO1 than in KO2. Other known SARS-CoV-2 entry factors were, however, not on the list. Those included alternative receptors such as transmembrane protein 106B (TMEM106B) [45], transferrin receptor [46], Basigin (CD147) [47], AXL receptor tyrosine kinase (AXL) [48], kringle containing transmembrane protein 1 (KREMEN1) and asialoglycoprotein receptor 1 (ASGR1) [49], attachment factors such as Neuropilin-1 [50], Niemann-Pick C1 (NPC1) [51], lectin receptors [52], as well as endosomal protease such as cathepsin L [53] (S2 Table). Interestingly, we noticed



**Fig 3. SAMHD1 depletion suppresses spike protein-mediated viral entry. (A)** Viral RNA levels (*N* and *E* genes) of Calu-3 Ctrl cells infected with ΔS-VRP(G) at 1.2 x 10⁵ copy number per cell and treated with 5 μM remdesivir (RDV) or DMSO were quantified with RT-qPCR at 24 hpi. 18S RNA was used as normalization control. Biological triplicate experiments were performed. **(B, C and D)** Calu-3 Ctrl, KO1 or KO2 cells were infected with

ΔS-VRP(G) at 1.2 x 10⁵ copy number per cell. **(B)** Viral RNA levels (*N* and *E* genes) were quantified with RT-qPCR at indicated time points. 18S RNA was used as normalization control. Biological triplicate experiments were performed. **(C)** Viral N protein was detected with Western blot at indicated time points. GAPDH was used as input control. The relative band intensities of viral N were calculated by dividing them with GAPDH bands and were normalized to Ctrl cells at 24 hpi. **(D)** The activities of secreted *Guassia* luciferase at indicated time points were assayed through Pierce *Gaussia* Luciferase Flash Assay Kit. Biological quadruplicate experiments were performed. **(E)** Cellular luciferase activities of HIV-1-spike-Luc/ZsGreen infected Calu-3 Ctrl, KO1 or KO2 cells (12.5 TCID$_{50}$ per cell) were measured at 48 hpi. Biological sextuplicate experiments were performed. For **(A)**, (B) and **(D)**, two-way ANOVA multiple comparisons test was performed. For **(E)**, one-way ANOVA multiple comparisons test was used. * P < 0.05, ** P < 0.01, *** P < 0.001. ns, not significant.

both KO1 and KO2 had enhanced expression of furin protease (Fig 4B), which may facilitate S protein priming in infected cells [54].

Finally, *ACE2* and *TMPRSS2* mRNA and protein expressions were validated respectively by RT-qPCR (Fig 4C) and Western blot (Figs 4D and S5A-S5C). For ACE2, we designed a primer pair targeting the full-length isoforms but not the N-terminally truncated ACE2 (dACE2) that is not a functional receptor to SARS-CoV-2 [55]. We employed a monoclonal ACE2 antibody that detected an N-terminal region (peptide sequence surrounding Asp201) that is absent in dACE2. For TMRPSS2, we designed a primer pair that can detect isoform 1–3. We employed a monoclonal TMPRSS2 antibody that detected the protease domain which is present in both zymogen (~ 55 kDa) and active form (~ 31 kDa).

We found that the full length *ACE2* mRNA and protein expressions were significantly reduced in KO1 and KO2 cells (Figs 4C, 4D, S5A and S5B). We confirmed the decreased full length *ACE2* mRNA expression in KO1 and KO2 by another qPCR primer pair [55] (S6A Fig). Interestingly, we found that KO2, but not KO1 cells, expressed significantly higher levels of dACE2 (S6B Fig), which is known to be induced by IFN-I [55,56]. In contrast, we found that *TMPRSS2* mRNA expressions were only downregulated in KO1 but not in KO2 cells (Fig 4C). Indeed, mRNA-seq results showed that TMPRSS2 transcripts were marginally defined as DEG in KO2 cells with log$_2$ fold change equal to -1.01 (Fig 4B, right panel). The total levels of TMPRSS2 protein (i.e., both zymogen and active form) trended to be reduced in KO1, although the difference was not statistically significant when compared with Ctrl cells (Figs 4D, S5A and S5C). Moreover, the TMPRSS2 protein in KO2 cells was not changed compared with Ctrl cells (Figs 4D, S5A and S5C). Therefore, only the expression of full-length ACE2, but not TMPRSS2 was confirmed to be commonly repressed in SAMHD1 KO Calu-3 cells. This suggested that SAMHD1 KO suppressed full length ACE2 expression, and thereby restricted S protein-mediated viral entry and SARS-CoV-2 infection.

## SAMHD1 KO does not affect *ACE2* mRNA stability

*ACE2* mRNA levels were downregulated in SAMHD1 KO cells (Figs 4B-4C and S6A). To test whether SAMHD1 KO affected *ACE2* mRNA stability, an actinomycin D (actD) treatment assay was performed as previously described [57,58]. Ctrl, KO1 and KO2 cells were treated with actD that inhibited transcription. At various time points following the treatment, the *ACE2* mRNA levels were measured by three specific qPCR primer pairs: the same qPCR primer pair used in Fig 4C for full length ACE2 targeting exon 2/3 junction and exon 4 (Fig 5A, primer pair 1 in S5 Table), a qPCR primer pair targeting the 5' end of *ACE2* mRNA at exon 1/2 junction (Fig 5B, primer pair 2 in S5 Table), and a qPCR primer pair targeting the 3' end of *ACE2* mRNA at exon 19 (Fig 5C, primer pair 3 in S5 Table). We found that *ACE2* mRNA was relatively stable over time that more than 50% of it remained by 8 hr of actD treatment (Fig 5A-5C), similar to the observation by others [59].

We included additional controls to validate the actD treatment assay. The mRNA stability of a proapoptotic gene BCL-2 interacting killer (BIK) is suppressed by La-related protein 1 (LARP1), a protein highly expressed in cancer cells [60,61]. We found that *BIK* mRNA in Calu-3 Ctrl cells exhibited rapid decay following actD treatment, with levels decreasing to below 50% by 4 hr and approximately 10% remaining at 8 hr (Fig 5D). Comparatively, 18S ribosomal RNA, which is a structural component of ribosome and more stable than mRNA, was found to remain above 50% of the initial levels during



**Fig 4. SAMHD1 depletion downregulates the expression of SARS-CoV-2 receptor ACE2. (A)** Calu-3 Ctrl, KO1 and KO2 cells were subjected to mRNA-seq using Element Aviti24. Biological triplicate was performed. DEGs were identified and selected based on the following criteria: FDR-adjusted P<0.01 and log$_2$ fold change >1. The number of DEGs in the three different comparison groups were plotted on a Venn diagram (i.e., Ctrl vs KO1, Ctrl

vs KO2 and KO1 vs KO2). 681 common DEGs of Ctrl vs KO1 and Ctrl vs KO2 were found to have consistent direction of change. **(B)** The 681 DEGs described in (A) were plotted on two volcano plots with FDR-adjusted P values against $\log_2$ fold change according to the values of KO1 vs Ctrl (left) or KO2 vs Ctrl (right). Three known viral entry factors ACE2, TMPRSS2 and Furin were identified. The values of $\log_2$ fold change were labelled adjacent to the corresponding data points. **(C)** mRNA levels of *ACE2* (primer pair 1, S5 Table) and *TMPRSS2* of Ctrl, KO1 or KO2 cells were quantified with RT-qPCR. 18S RNA was used as normalization control. Biological sextuplicate experiments were performed. **(D)** Protein levels of ACE2 and TMPRSS2 were measured with Western blot. GAPDH was used as the input control. Triangle (◁) and asterisk (*) symbols respectively represented the zymogen (~55 kDa) and active form (~31 kDa) of TMPRSS2. The relative band intensities of ACE2, TMPRSS2 zymogen and TMPRSS2 active form were calculated by dividing them with GAPDH bands and were normalized to Ctrl. For **(C)**, one-way ANOVA multiple comparisons test was used to evaluate the statistical significance of the difference between Ctrl and KO1 or KO2. **** $P < 0.0001$, ns, not significant.

the actD treatment assay (Fig 5E). These results confirmed that the actD treatment assay was valid in distinguishing unstable RNA species. We found that *ACE2* mRNA in actD-treated KO1 or KO2 cells were similarly stable as compared with that of the Ctrl cells with over 50% of *ACE2* mRNA remaining in both KO1 and KO2 cells at 8 hr actD treatment (Fig 5A-5C). Therefore, SAMHD1 KO did not affect *ACE2* mRNA stability.

## SAMHD1 KO downregulates the expressions of HNF1α, HNF1β and HNF4α in Calu-3 cells

Next, we questioned how SAMHD1 KO affected *ACE2* transcription. The *ACE2* promoter is a bipartite promoter that is controlled by various transcription factors depending on cell types [62]. Analysis of human lung tissues and immortalized lung 16HBE cells revealed STAT3 transcriptionally controlled ACE2 expression [63]. HNF1α and HNF1β were found to cooperate and promote ACE2 expression transcriptionally in HEK293 and pancreatic islet cells [64,65]. HNF4α was found to be a transcription repressor or activator of ACE2 expression depending on different cell types and experiment settings [66–68]. Specificity protein 1 (Sp1) was found to transcriptionally promote ACE2 expression in immortalized human type II alveolar epithelial cells [68]. Forkhead box A2 (FOXA2) was found to promote ACE2 transcription in 832/13 insulinoma cells or mouse pancreatic islets [69]. Ikaros was found to drive ACE2 expression in human cardiac fibroblasts in response to angiotensin II stimulation by binding to ATTTGGA sequence of the proximal promoter [70].

The exact transcription factors regulating endogenous ACE2 expression in Calu-3 cells remain unclear. To the best of our knowledge, GATA-binding protein 6 (GATA6) was the only experimentally identified transcription factor in Calu-3 cells promoting ACE2 expression and facilitating SARS-CoV-2 infection [71]. GATA6 was identified by CRISPR/Cas9-KO library screening for surviving cells upon SARS-CoV-2 infection. Two additional independent studies identified HNF1β as a significant candidate to reduce surviving cells upon SARS-CoV-2 infection by CRISPR/dCas9-activation library screening [72,73]. However, unlike GATA6, the role of HNF1β in regulating ACE2 expression in Calu-3 cells was not further validated. Other known ACE2 transcription factors as mentioned above were not on the candidate list of the three screening studies [71–73]. Our RT-qPCR analysis revealed that the mRNA levels of *HNF1β* were reduced in KO1 and KO2 cells (Fig 6B), while *GATA6* mRNA level was unchanged in KO1 and KO2 cells compared to Ctrl cells (Fig 6D).

HNF1α, HNF1β, and HNF4α are core transcription factors that establish a complex and interdependent regulatory network during liver and pancreas development [74,75]. Particularly in hepatocytes, HNF1α and HNF1β can transcriptionally promote the expression of HNF4α [76]. In return, HNF4α promotes HNF1α's expression [77,78] and transcriptional activity [79]. This network in lung cells is unclear. We found that the mRNA levels of *HNF1α* (Fig 6A) and *HNF4α* (Fig 6C) were also downregulated in KO1 and KO2 cells. Western blot analysis showed that the protein levels of HNF1α, HNF1β and HNF4α were reduced in KO1 and KO2 cells compared to Ctrl cells (Figs 6E, S5A, and S5D-S5F). These results suggested that SAMHD1 KO suppressed the expression of the HNF1α, HNF1β and HNF4α in Calu-3 cells.

## Knocking down HNF1α and HNF1β inhibits endogenous ACE2 expression in Calu-3 cells

Whether HNF1α, HNF1β and HNF4α controlled the endogenous ACE2 expression in Calu-3 cells was unclear. To address this question, HNF1α, HNF1β and HNF4α were knocked down individually in Calu-3 cells. Dicer-substrate siRNAs



Fig 5. SAMHD1 KO does not affect *ACE2* mRNA stability. (A-C) The levels of *ACE2* mRNA of actinomycin D (10 µg/mL) treated Calu-3 Ctrl, KO1 or KO2 cells were quantified with RT-qPCR using (A) primer pair 1, (B) primer pair 2 or (C) primer pair 3 (S5 Table) at indicated time points. (D) *BIK* mRNA and (E) 18S RNA were detected with RT-qPCR as controls to the experiment. Total RNA (1 µg) was used for reverse transcription. mRNA levels were normalized to 0 hr. The one phase decay curves for KO1 (blue), KO2 (red) and Ctrl cells (black) were plotted.

(DsiRNA) transfection efficiently reduced the target gene expression by more than 50% (Figs 7A-7C and S7A). We noticed that knocking down HNF1β, but not HNF1α, suppressed *HNF4α* mRNA and protein expression in Calu-3 cells. Knocking down HNF4α suppressed *HNF1α* mRNA but not protein (Fig 7B and 7C). Regarding endogenous ACE2 expression, we found that knocking down HNF1α and HNF1β, but not HNF4α, significantly suppressed ACE2 protein expression (Figs 7A-7B and S7A). Interestingly, only knocking down HNF1α led to a significant reduction of *ACE2* mRNA expression (Fig 7C). Knocking down HNF1β suppressed ACE2 protein expression (Figs 7A-7B and S7A Fig) but not mRNA expression in Calu-3 cells (Fig 7C). Finally, we found that knocking down either HNF1α or HNF1β significantly suppressed SARS-CoV-2 infection in Calu-3 cells (Fig 7D). Therefore, HNF1α and HNF1β were required to promote the expression of ACE2 and support SARS-CoV-2 infection in Calu-3 cells.

## HNF1-dependent ACE2 expression is suppressed in SAMHD1 KO Calu-3 cells

We hypothesized that the functions of HNF1α and HNF1β in promoting endogenous ACE2 expression were suppressed in the SAMHD1 KO Calu-3 cells. To test the hypothesis, transient HNF1α and HNF1β knockdown was performed in SAMHD1 KO cells to examine if the ACE2 expression was less dependent on HNF1α and HNF1β when their expression

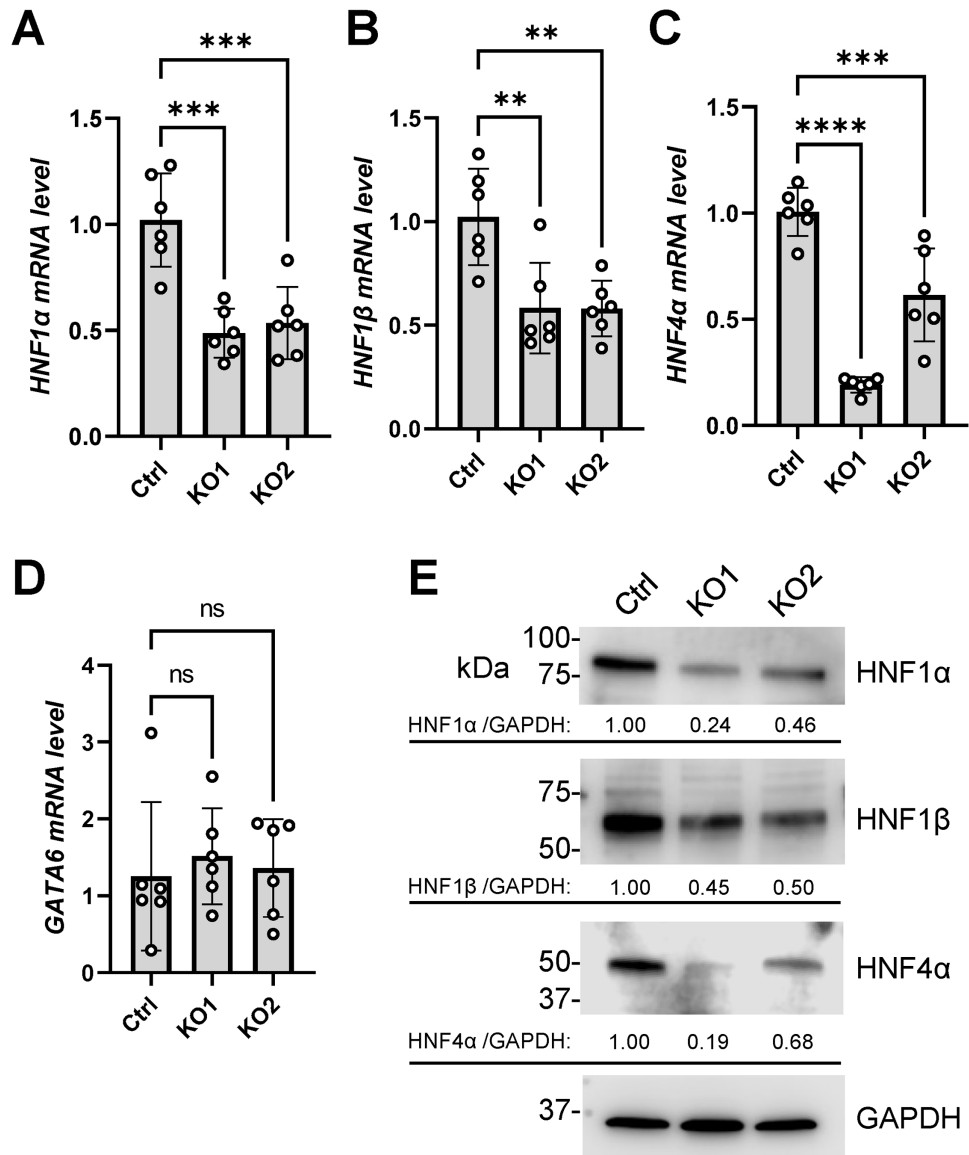

**Fig 6. SAMHD1 KO decreases the expression HNF1α, HNF1β and HNF4α.** mRNA levels of **(A)** *HNF1α*, **(B)** *HNF1β*, **(C)** *HNF4α* and **(D)** *GATA6* were measured with RT-qPCR. 18S RNA was used as normalization control. **(E)** Protein levels of HNF1α, HNF1β and HNF4α were detected with Western blot. GAPDH was detected for input control. The relative band intensities of HNF1α, HNF1β and HNF4α were calculated by dividing them with GAPDH bands and were normalized to Ctrl cells. For **(A)**-(D), one-way ANOVA multiple comparisons test was used to evaluate the statistical significance of the difference between Ctrl and KO1 or KO2. ** $P<0.01$, *** $P<0.001$, **** $P<0.0001$. ns, not significant.

was suppressed upon SAMHD1 deficiency. We found that HNF1α and HNF1β protein expression in KO1 and KO2 cells could still be suppressed by DsiRNA knockdown despite their lower expression levels compared to Ctrl cells (Figs 8A-8C and S7B Fig). Knocking down HNF1α and HNF1β significantly reduced ACE2 protein expression in Ctrl cells (Figs 8A, 8D and S7B Fig). In the case of KO1 and KO2 cells, the suppression of ACE2 protein expression by HNF1α and HNF1β knockdown was mitigated. Knocking down HNF1α, but not HNF1β, reduced *ACE2* mRNA expression in Ctrl cells (Fig 8E). In contrast, HNF1α knockdown was less potent in further suppressing *ACE2* mRNA in KO1 and KO2 cells (Fig 8E). Given



**Fig 7. Knocking down HNF1α and HNF1β downregulates ACE2 expression and suppresses SARS-CoV-2 infection. (A)** ACE2, HNF1α, HNF1β or HNF4α proteins of Calu-3 cells (4 x 10⁵) treated with DsiRNA (24 pmol, 72 hpt) targeting HNF1α, HNF1β, HNF4α or NC were detected with Western blot. GAPDH was detected for input control. A representative blot was shown. **(B)** ACE2, HNF1α, HNF1β or HNF4α protein band intensities of three independent experiments (i.e., (A) and Supplementary S7A) were quantified by densitometry analysis and divided by GAPDH and normalized to NC control. **(C)**

Cells were treated similarly as in **(A)**. The mRNA levels of *ACE2* (primer pair 1, S5 Table), *HNF1α*, *HNF1β* or *HNF4α* were quantified with RT-qPCR. 18S RNA was used as normalization control. **(D)** Calu-3 cells (3.2 x 10⁶) transfected with DsiRNA (192 pmol, 72 hpt) targeting HNF1α, HNF1β or NC were infected with authentic SARS-CoV-2 at MOI = 0.05. Released viral progenies in culture supernatants were quantified with plaque assay. Biological triplicate experiments were performed. For **(B-C)**, unpaired T-test was used to evaluate the statistical significance of the difference between sample groups and NC. For **(D)**, two-way ANOVA multiple comparisons test was performed comparing Ctrl and KO1 or KO2 at various time points. * $P < 0.05$, ** $P < 0.01$, *** $P < 0.001$, **** $P < 0.0001$. ns, not significant.

that endogenous ACE2 expression was more dependent on HNF1α and HNF1β in Calu-3 Ctrl cells (Fig 8D and 8E), our data suggested that SAMHD1 KO inhibited HNF1-dependent ACE2 expression in Calu-3 cells.

### SAMHD1 knock-in (KI) promotes *ACE2* mRNA but not protein expression or SARS-CoV-2 infection

To further delineate the mechanisms by which SAMHD1 KO suppressed ACE2 expression and SARS-CoV-2 infection, SAMHD1 expression was reintroduced into KO cells via a lentiviral vector that expressed sgRNA-1/2 resistant SAMHD1. The transduced stable cell lines were selected and maintained through hygromycin B containing media. We only obtained stable SAMHD1 KI cells with the KO2 clone, but not the KO1 clone due to significant cell death during the selection. The resulting cell line was named KO2/KI. For mock control, KO2 cells were transduced with the empty lentiviral vector counterpart, and the resulting cell line was named KO2/vec.

We found that KO2/KI had significantly higher *SAMHD1* mRNA and protein levels than KO2/vec, indicating successful reintroduction of SAMHD1 expression (Fig 9A and 9B). We noticed that the proinflammatory cytokine *IL-6* mRNA expression levels of KO2/vec and KO2/KI cells were around 10-fold higher than that of the parental KO2 cells, which was already higher than that of the Ctrl cells (Figs 1C and 9A). Reintroducing SAMHD1 expression in KO2/KI cells upregulated *ACE2* mRNA levels when compared with KO2/vec but did not affect *IL-6* mRNA expression (Fig 9A). Comparatively, SAMHD1 re-expression in KO2 cells did not rescue ACE2 protein expression (Fig 9B and 9C) or SARS-CoV-2 infection (Fig 9D and 9E). We found that KO2/KI cells had higher HNF1α but similar HNF1β protein expression levels when compared with KO2/vec cells (Fig 9B and 9C). This indicated that SAMHD1 reintroduction in KO2 cells only reconstituted HNF1α but not HNF1β protein expression. Thus, the incomplete HNF1 reconstitution promoted *ACE2* mRNA expression yet failed to increase ACE2 protein levels or affect SARS-CoV-2 infection.

### Elevated dNTP levels suppress SARS-CoV-2 infection and ACE2 expression

Given the dNTPase activity of SAMHD1 in cells, we asked if dNTP level contributed to modulating SARS-CoV-2 infection and ACE2 expression in Calu-3 cells. Calu-3 cells were supplemented with the four deoxynucleosides (dN) which promoted dNTP salvage pathway and dNTP levels as previously described [80]. We confirmed that dN supplementation promoted dATP and dTTP levels of Calu-3 cells at 14 hr post-treatment (Fig 10A). In contrast, dGTP and dCTP levels were insignificantly changed by dN supplementation, possibly because SAMHD1 has a better rate of hydrolysis to dGTP and dCTP than dATP and dTTP [81].

Intriguingly, we found that dN supplementation inhibited SARS-CoV-2 replication as demonstrated by a reduction of infectious viral progeny (Fig 10B) and intracellular viral RNA expression (Fig 10C and 10D). Moreover, dN supplementation to cells suppressed ACE2 protein expression by ~30% (Fig 10E and 10F). dN supplementation also trended to suppress *ACE2* mRNA expression although the change was not statistically significant (Fig 10G). Moreover, dN supplementation promoted TMPRSS2 protein expression but had no effect on *TMPRSS2* mRNA expression (Fig 10E-10G). We also found that dN supplementation reduced the expression of HNF1β by ~20% but had no effect to the expression of HNF1α (Fig 10H and 10I). dN supplementation had no effect on SAMHD1 expression (Fig 10H and 10I) or proinflammatory cytokine *IL-6* mRNA expression (Fig 10G). These results suggested that enhanced dNTP levels suppressed SARS-CoV-2 infection by downregulating the expression of ACE2 protein without affecting SAMHD1 expression or proinflammatory cytokine expression.



**Fig 8. SAMHD1 deficiency mitigates the effect of HNF1α and HNF1β knockdown in further suppressing ACE2 expression. (A)** ACE2, HNF1α and HNF1β proteins of Ctrl, KO1 or KO2 cells (2 x 10⁵) treated with DsiRNA (12 pmol, 72 hpt) targeting HNF1α, HNF1β or NC were detected with Western blot. GAPDH was detected for input control. A representative blot was shown. **(B)** HNF1α, **(C)** HNF1β and **(D)** ACE2 protein band intensities of three



independent experiments (i.e., (A) and Supplementary S4B) were quantified by densitometry analysis, divided by GAPDH bands and normalized to Ctrl NC. **(E)** Cells were treated similarly as in **(A)**. The mRNA levels of *ACE2* (primer pair 1, S5 Table) were quantified with RT-qPCR. 18S RNA was used as normalization control. One-way ANOVA multiple comparisons test was used to evaluate the statistical significance of the difference between sample groups. * $P < 0.05$, ** $P < 0.01$, *** $P < 0.001$, **** $P < 0.0001$. ns, not significant.

## Discussion

In this study, we investigated the role of SAMHD1 in SARS-CoV-2 infection of Calu-3 cells, which expressed endogenous ACE2 and supported productive viral replication. Similar to our previous study using other cell types [32], SARS-CoV-2 replication was hindered in SAMHD1 KO Calu-3 cells. Previous study showed that inhibition of IFN signaling by the JAK1/2 inhibitor baricitinib alleviated the replication restraint of SARS-CoV-2 in HEK293T cells with SAMHD1 KO [32], while in the current study, baricitinib treatment did not promote SARS-CoV-2 replication in SAMHD1 KO Calu-3 cells. We noticed that baricitinib treatment promoted the intracellular viral N RNA and protein levels of SARS-CoV-2 in Calu-3 Ctrl cells but did not change the production of infectious progeny viruses. In contrast, baricitinib treatment enhanced neither the viral RNA nor viral protein levels in SAMHD1 KO1 and KO2 cells, suggesting that some fundamental mechanisms to SARS-CoV-2 replication were disrupted in KO1 and KO2 cells rather than IFN signaling.

HEK293T cells support productive replication of SARS-CoV-2 despite negligible expression of ACE2 and TMPRSS2 [82–84]. Through a CRISPR/Cas9 KO library screen, an alternative viral receptor TMEM106B was found to be required for SARS-CoV-2 replication in HEK293T cells [84]. In contrast, SARS-CoV-2 infection in Calu-3 cells exhibits a strong dependence on ACE2 expression, as evidenced by multiple CRISPR/Cas9 library screens where ACE2 was consistently the most critical host factor identified [71–73]. We found that compared to other known viral entry factors, ACE2 expression was significantly reduced in SAMHD1 KO Calu-3 cells, with consistent results observed in both the KO1 and KO2 lines. SAMHD1 KO significantly hindered SARS-CoV-2 S protein-mediated viral entry in Calu-3 cells; however, once this barrier was bypassed by VSV-G pseudotyping, the single-cycle infection of SARS-CoV-2 replicon was restored. Therefore, SAMHD1 KO abolished ACE2-dependent and S-mediated viral entry which was required for SARS-CoV-2 infection of Calu-3 cells, but not HEK293T cells.

We questioned how SAMHD1 KO suppressed ACE2 expression in Calu-3 cells. We found that SAMHD1 KO repressed *ACE2* mRNA expression in Calu-3 cells, but *ACE2* mRNA stability was unaffected. This finding suggested that SAMHD1 KO pre-transcriptionally or transcriptionally suppressed *ACE2* mRNA expression. We found that SAMHD1 KO suppressed the expression levels of HNF1α and HNF1β, which in turn downregulated the expression of ACE2 and SARS-CoV-2 infection. We functionally validated the inhibition of HNF1-mediated endogenous ACE2 expression in SAMHD1 KO Calu-3 cells. To the best of our scope, this is the first study functionally confirming the role of HNF1α and HNF1β in promoting ACE2 expression in lung epithelial cells. Interestingly, although HNF1α and HNF1β were previously defined to be transcription factors of ACE2 by binding to the HNF1 response element of the proximal promoter [64,65], we found that only HNF1α was required for *ACE2* mRNA expression in Calu-3 cells. HNF1β was however required for ACE2 protein but not mRNA expression. These findings implied that SAMHD1 KO contributed to suppress *ACE2* mRNA and protein levels through downregulation of HNF1α and HNF1β expression, respectively (summarized mechanisms in Fig 11).

It is still in question how SAMHD1 KO represses the expression and activity of HNF1α and HNF1β in driving ACE2 expression. SAMHD1 is a nuclear protein that has not been identified to have direct transcriptional function. SAMHD1 mainly interacts with single-stranded RNA or DNA but had low affinity to double-stranded DNA [85,86]. One possibility is that SAMHD1 depletion may have created some cellular stresses that inhibit the expression of HNF1α and HNF1β. We found that SAMHD1 KO Calu-3 cells spontaneously expressed higher IL-6 than Ctrl cells. Indeed, the expression of several proinflammatory cytokines including IL-1α, IL-1β, colony-stimulating factor 2 (CSF2) and IL-11 were also upregulated in both KO1 and KO2 clones (S2 Table). Some studies suggested that HNF1α and HNF1β expressions were suppressed

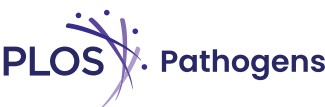

**Fig 9. Reintroducing SAMHD1 expression in KO2 cells promotes *ACE2* mRNA but not protein expression. (A)** Levels of *SAMHD1*, *ACE2* (primer pair 1, S5 Table) and *IL-6* mRNA were measured with RT-qPCR. 18S RNA was used as internal control. Three to six biological replicates were performed. **(B)** ACE2, HNF1α, HNF1β and SAMHD1 proteins of KO2/vec or KO2/KI were detected with Western blot. GAPDH was detected for input



control. Three independent experimental replicates were performed. **(C)** ACE2, HNF1α and HNF1β protein band intensities of three independent experiments shown in Fig 9B were quantified by densitometry analysis, divided by GAPDH and normalized to KO2/vec, **(D and E)** KO2/vec and KO2/KI cells were infected with authentic SARS-CoV-2 at MOI = 1. At 24 hpi, (D) released viral progenies in culture supernatants were quantified with plaque assay and (E) intracellular viral RNA was quantified with RT-qPCR detecting viral N gene and normalized to 18S rRNA. Unpaired T-test was used to evaluate the statistical significance of the difference between sample groups. * P < 0.05, ** P < 0.01, *** P < 0.001, **** P < 0.0001. ns, not significant.

upon inflammation. Firstly, the expression of HNF1α in hepatocytes was suppressed through a positive feedback circuit when mice were treated with dimethylnitrosamine that induced liver inflammation followed by hepatic fibrogenesis [87]. Secondly, the expression of HNF1β was downregulated by IFN-γ in kidney cells HK-2, while lipopolysaccharide (LPS) injection downregulated HNF1β expression in mouse kidney cells [88]. Thirdly, we found that HNF4α knockdown reduced *HNF1α* mRNA expression in Calu-3 cells although the protein change was not observed (Fig 7A-7C). One recent study showed that HNF4α was transcriptionally suppressed by inflammatory stimuli such as IL-6 and IL-1β in differentiated HepaRG cells [89]. It is possible that the enhanced expression of proinflammatory cytokines of SAMHD1 KO Calu-3 cells suppresses HNF4α expression, thereby decreasing the expression of HNF1α. We also found that HNF1β positively regulated HNF4α expression (Fig 7B-7C). It is possible that HNF1β plays an upstream role in the expression of HNF1α and HNF4α in Calu-3 cells. SAMHD1 expression prevents spontaneous proinflammatory cytokine production mainly through inhibition of NF-kB signaling [25,28] and preventing RIG-I like receptor signaling and cGAS-STING activation by self-RNA or self-DNA species [17–24]. Therefore, SAMHD1 expression maintains innate immune homeostasis, which sustains the endogenous expressions of HNF1α and HNF1β to support ACE2 expression.

SAMHD1 KO promoted cellular dNTP levels (S1C Fig). Strikingly, without changing the expression levels of SAMHD1, increase of cellular dNTP through dN supplementation was sufficient to recapitulate some phenotypes of SAMHD1 KO Calu-3 cells including inhibition to SARS-CoV-2 infection and ACE2 protein expression (Fig 10B-10F). We found that dN supplementation suppressed HNF1β expression, which likely contributed, at least in part, to the decreased ACE2 expression (Fig 10E-10I). HNF1α expression was however unaffected by dN supplementation. Conversely, reintroduction of SAMHD1 expression in KO2 cells promoted HNF1α expression but not HNF1β (Fig 9B and 9C).

It was worth noting that SAMHD1 KI Calu-3 cell lines KO2/vec and KO2/KI exhibited higher *IL-6* mRNA expression levels than the parental KO2 cells (Fig 9A). Possibly, the transduced lentiviral vectors elicited innate immune responses and enhanced proinflammatory cytokine production [90]. Unexpectedly, SAMHD1 expression in KO2/KI cells did not modulate the levels of IL-6 expression (Fig 9A), which was different from our previous studies showing that SAMHD1 KI suppressed LPS- or virus-induced IL-6 expression in SAMHD1-defective monocytic THP-1 or U937 cells [25,40]. This difference was likely attributable to variations in cell type and experimental conditions such as the approaches in isolating transduced cells and/or the use of different selection antibiotics [40,91]. At elevated proinflammatory cytokine levels, HNF1α expression was upregulated by SAMHD1 KI. Given that dN supplementation did not affect HNF1α expression, this indicated that SAMHD1 promoted HNF1α expression independent to proinflammatory cytokine expression and dNTP levels. HNF1β expression was however dependent to cellular dNTP level as demonstrated by the dN supplementation assay (Fig 10H and 10I). Interestingly, HNF1β expression was not upregulated by SAMHD1 KI which should suppress dNTP [25,40]. Perhaps the elevated proinflammatory cytokine levels of the SAMHD1 KI cell lines repressed HNF1β expression. Overall, these results suggested that HNF1β expression was dependent to cellular stresses elicited by SAMHD1 KO, such as proinflammatory cytokines and dNTPs, whereas HNF1α expression was regulated by SAMHD1 protein independently of these stresses (Fig 11). Further study is needed to clarify exactly how SAMHD1 depletion downregulates HNF1α and HNF1β expression to inhibit ACE2 expression.

We noticed that several cell surface proteins and proteins related to extracellular matrix distinguished SAMHD1 KO Calu-3 cells from Ctrl cells through the PCA (S4C Fig). Similarly, gene ontology (GO) enrichment analysis (GOEA) of cellular components to the 681 common DEGs of SAMHD1 KO Calu-3 cells suggest that cell periphery was the best enriched



**Fig 10. Increasing dNTP levels suppresses ACE2 expression and SARS-CoV-2 infection.** (A) dATP, dTTP, dGTP and dCTP levels of Calu-3 cells treated with DMEM or dN for 14 hr were measured with HIV-1 RT based single nucleotide incorporation assay. (B) Calu-3 cells treated with DMEM or dN were infected with authentic SARS-CoV-2 at MOI = 1. At 24 hpi, (B) released viral progenies in culture supernatants were quantified with plaque assay

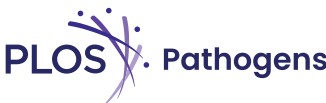

and (C-D) intracellular viral RNA was quantified with RT-qPCR detecting (C) viral N gene or (D) viral RdRP, which were normalized to 18S rRNA. (E) ACE2 and TMPRSS2 proteins of DMEM (-) or dN (+) treated Calu-3 cells were detected with Western blot. GAPDH was detected for input control. Triangle (◁) and asterisk (*) symbols respectively represented the zymogen (~55 kDa) and active form (~31 kDa) of TMPRSS2. Three independent experimental replicates were performed. (F) ACE2 and TMPRSS2 (◁+*) protein band intensities of three independent experiments shown in Fig 10E were quantified by densitometry analysis, divided by GAPDH and normalized to DMEM control. (G) Levels of *ACE2* (primer pair 1, S5 Table), *TMPRSS2* and *IL-6* mRNA were measured with RT-qPCR. 18S RNA was used as internal control. Three biological replicates were performed. (H) HNF1α, HNF1β and SAMHD1 proteins of DMEM (-) or dN (+) treated Calu-3 cells were detected with Western blot. GAPDH was detected for input control. (I) HNF1α, HNF1β and SAMHD1 protein band intensities of three independent experiments shown in Fig 10H were quantified by densitometry analysis, divided by GAPDH and normalized to DMEM control. Unpaired T-test was used to evaluate the statistical significance of the difference between sample groups. * P<0.05, ** P<0.01, *** P<0.001. ns, not significant.

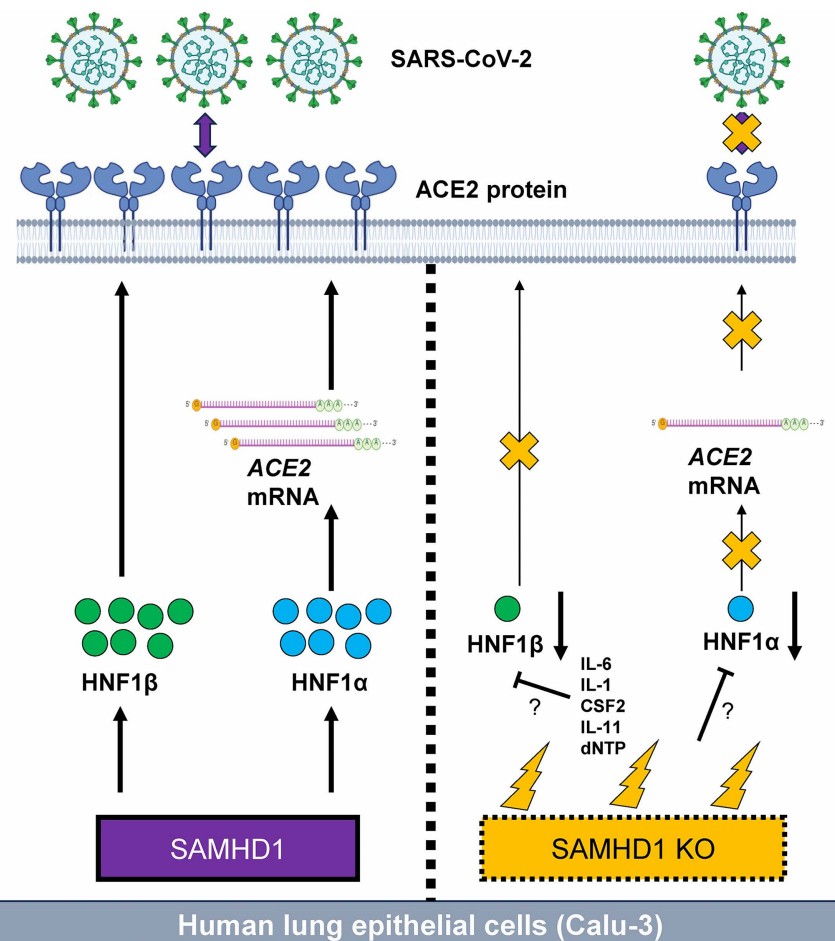

**Fig 11. SAMHD1 KO restricts SARS-CoV-2 infection via suppression of HNF1-mediated ACE2 expression.** At normal condition, SAMHD1 supports the expression of HNF1α and HNF1β which respectively promote mRNA and protein expression of ACE2 in human lung epithelial cells (Calu-3). Upon infection, SARS-CoV-2 binds to ACE2 receptor via its surface spike protein to initiate viral entry. In the absence of SAMHD1 expression (i.e., SAMHD1 KO), cellular stresses are induced such as proinflammatory cytokine expression (e.g., IL-6, IL-1, CSF2, or IL-11) and elevated dNTP levels (thunder symbols). The expression levels of HNF1α and HNF1β are reduced. The expression of HNF1β is more dependent to the stresses induced by SAMHD1 KO whereas the expression of HNF1α is less dependent to the stresses but more to the level of SAMHD1 protein expression (possibly other SAMHD1-dependent factors). In turn, ACE2 expression is reduced which blocks SARS-CoV-2 infection via restricting spike protein-mediated viral entry. Created in BioRender. Wu, **L.** (2026) https://BioRender.com/gjkpjjj.

                                                  21 / 33

term having the lowest FDR-adjusted P value (S8 Fig and S3 Table). Among the top ten GO terms, four terms were correlated with cell surface membranes in addition to cell periphery (including plasma membrane, apical plasma membrane and cell surface) while five terms were correlated with extracellular matrix (including extracellular matrix, external encapsulating structure, extracellular region, extracellular space, collagen-containing extracellular matrix). Although we did not observe any validated SARS-CoV-2 entry factors on the 681 common DEGs except ACE2, TMPRSS2 and furin in which only ACE2 was found to be consistently downregulated in both KO1 and KO2 clones, it might be possible that the changes in cell surface and extracellular matrix structures contribute to the suppressed SARS-CoV-2 infection and viral entry in SAMHD1 KO Calu-3 cells. Any combinatorial effects of the DEGs contributing to the changes in cell surface and extracellular structures that may potentially suppress SARS-CoV-2 infection shall be valuable in future investigation (S4 Table).

Why SAMHD1 depletion affected the expression of so many genes encoding cell surface and extracellular matrix proteins remains unclear. One study suggests SAMHD1 expression promotes focal adhesion kinase (FAK) signaling by binding to cortactin, and in turn activated Rac family small GTPase 1 (Rac1)-mediated lamellipodia formation in human clear cell renal cell carcinoma [92]. It is possible that SAMHD1 KO Calu-3 cells had lower activity of FAK and Rac-1. Interestingly, Rac-1 was found to promote SARS-CoV-2 entry via macropinocytosis in ACE2-expressing cells [93]. Thus, SAMHD1 may enhance SARS-CoV-2 infection by macropinocytosis through cortactin, FAK and Rac-1, in addition to promoting HNF1-mediated ACE2 expression. Further investigation is required for the detailed mechanism by which SAMHD1 promotes SARS-CoV-2 infection in physiologically relevant cell types.

We were surprised to find that only KO2 but not KO1 clone had spontaneous IFN-I responses (Fig 1C). Indeed, based on PCA, KO2 cells were clustered away from Ctrl and KO1 cells mainly by several ISGs (S4C Fig). IFN-I production requires activation of IRF3/7 in addition to NF-κB signal pathway [94]. KO1 cells exhibited complete absence of SAMHD1 protein since expression was abolished with sgRNA1 frameshifting exon 1 at [11]Lysine (Figs 1A, 1B and S1A). KO2 cells may express a truncated protein consisting of the first 1–149 amino acids of SAMHD1. Whether the truncated SAMHD1 protein (aa. 1–149) contributes to IFN-I activation and ISG expression in Calu-3 cells is unclear and will require further investigation.

In summary, we found that SAMHD1 depletion restricted SARS-CoV-2 infection in Calu-3 cells by repressing S protein-mediated viral entry via HNF1-mediated ACE2 expression. Our study shed light on novel functions and mechanisms of SAMHD1 in lung epithelial cells to facilitate SARS-CoV-2 infection.

## Materials and methods

### Cell culture

Cell culture was performed as previously described [91,95]. The original wildtype Calu-3 cells were obtained from ATCC (HTB-55). Wild-type Calu-3 cells as well as Ctrl, KO1 and KO2 cells were maintained in DMEM/F12 (ThermoFisher Scientific, Cat. no. 11320033) with 20% FBS, 100 U/mL penicillin (Gibco, Cat. no. 15140122), and 100 µg/mL streptomycin (Gibco, Cat. no. 15140122). Huh7.5 cells were kindly provided by Dr. Balaji Manicassamy (University of Iowa) [41]. Vero-hTMPRSS2 cells were cultured in DMEM (ThermoFisher Scientific, Cat. no. 11965092) with 10% FBS, 100 U/mL penicillin, 100 µg/mL streptomycin and 5 µg/mL blasticidin (Gibco, Cat. no. A1113903) as previously described [96]. Huh7.5 cells were cultured in DMEM with 10% FBS, 100 U/mL penicillin, 100 µg/mL streptomycin. ActD (SellekChemicals, Cat. no. S8964), remdesivir (MCE, Cat. no.: HY-104077) and baricitinib (MCE, Cat. no.: HY-15315) were dissolved in dimethyl sulfoxide (DMSO) and stored at -20°C.

### Generation of SAMHD1 KO Calu-3 cells

SAMHD1 KO Calu-3 cells were generated similarly as previously described with some modifications [91,95]. Briefly, third generation lentiviral particles were rescued from HEK293T cells (1 x 10$^7$) transfected with lentiCRISPR v2-puro empty vector (Addgene, plasmid no. 98290) lentiCRISPR v2-puro-sgRNA1, or lentiCRISPR v2-puro-sgRNA2 together

with psPAX2, and pMD2.G (ratio 4:3:1) by PEI (1:5 ratio). The target sequence of sgRNA 1 guided SpCas9 cutting at exon 1 of SAMHD1 gene (chr20: 36,951,611/36,951,612) while that of sgRNA 2 targeted SpCas9 for exon 4 (chr20: 36,935,087/36,935,088). The template sequences for sgRNA1 and sgRNA2 were listed in S5 Table. At 24 hr post-transfection (hpt), the culture supernatant was replenished with fresh media. At 72 hpt, the 12 mL culture supernatant was cleared by 500 x g 5 min centrifugation followed with 0.45 µm filtering and concentrated by lenti-concentrator (ORIGENE, Cat. no TR30026) into 1mL media. A day before transduction, 4 x 10⁵ Calu-3 cells were seeded in 12 well plates. On the day of transduction, the culture supernatant of the Calu-3 cells was replaced with the 1mL concentrated virus. At 24 hpi, the culture supernatant was replaced with fresh DMEM/F12 with 20% FBS, 100 U/mL penicillin, and 100 µg/mL strepto-mycin. At 72 hpi, cells were selected against 1 µg/mL puromycin. The survival cells of the transduced populations were purified by limited dilution on a 96 well plate. Finally, the purified cell clones were verified for expression of SAMHD1 and genotyping.

### Reintroducing SAMHD1 expression for KO2 cells

SAMHD1 KI cell lines KO2/vec and KO2/KI were produced similarly as previously described with some modifications [40,91]. sgRNA-1/2 resistant SAMHD1 was amplified and generated through overlapping PCR as mentioned before using Q5 High-Fidelity 2X Master Mix (NEB, Cat. no. M0492S) [97]. The sequences of the primers used in the first and second round PCR were listed in S5 Table. The resulting sgRNA-1/2 resistant SAMHD1 fragment was subcloned onto pLenti-CMV-hygro through BamHI and SalI restriction sites [98]. The resulting pLenti-CMV-hygro-SAMHD1-resist plasmid was used to produce lentiviral vector expressing SAMHD1. Similar to that described in the above section "Generation of SAMHD1 KO Calu-3 cells", KO2 cells were transduced with lentiviral vectors generated by either pLenti-CMV-hygro or pLenti-CMV-hygro-SAMHD1-resist, and then selected and maintained in DMEM/F12 with 20% FBS, 100 U/mL penicillin, and 100 µg/mL streptomycin with 350 µg/mL hygromycin B. The resulting stable cell lines respectively named KO2/vec or KO2/KI were used for subsequent studies.

### RNA interference and transfection

DsiRNAs were oligo-duplex and the sequences were predesigned by Integrated DNA Technologies (IDT) [99]. Cells were transfected with DsiRNA with Lipofectamine RNAiMAX Reagent (ThermoFisher Scientific, Cat. no. 13778150). In brief, equal volume of OPTIMEM- (Gibco, Cat. no. 31985070) diluted DsiRNA (150 µl for 24 well format and 300 µl for 12 well format) and OPTIMEM-diluted lipofectamine RNAiMAX (1:3 ratio) were mixed and incubated for 20 min at room tempera-ture. Cell suspension (0.5 mL in 24 well format and 1 mL in 12 well format) was mixed with the DsiRNA/lipofectamine RNAiMax complex. At 24 hpt, culture supernatant was replaced with fresh medium. Cells were harvested at indicated time points. The DsiRNA sequences used in the current study were listed in S5 Table.

### Virus stocks

Authentic SARS-CoV-2 (Wuhan-Hu-1) was rescued from Vero-hTMPRSS2 transfected with bacterial artificial chro-mosome carrying SARS-CoV-2 genome (BAC-SARS-CoV-2) as previously described with some modifications [100]. In brief, 1 x 10⁷ Vero-hTMPRSS2 cells were seeded in a T75 flask a day before transfection without blasticidin. Vero-hTMPRSS2 cells were transfected with 10 µg BAC-SARS-CoV-2 with lipofectamine 3000 (ThermoFisher Scientific, Cat. no. L3000008). At 24 hpi, the culture supernatant was replaced with serum-free DMEM. At 72 hpi, 80% cytopathic effect was observed. The crude culture supernatant was collected for one freeze-thaw cycle. Then, the supernatant was cleared by pelleting debris at 2,000 x g, 10 min, 4°C. The viral containing supernatant was aliquoted and frozen at -80°C for subsequent studies.

ΔS-VRP(G) was rescued from Huh7.5 transfected with BAC carrying spike-defective SARS-CoV-2 genome (BAC-ΔS-Gluc-T2A-nGFP) together with VSV-G expression plasmid similarly as previously described [41,101].

HIV-1-spike-Luc/ZsGreen (BEI resources, Cat. no. NR-53818) was a generous gift donated by Dr. Jesse Bloom (Fred Hutchinson Cancer Center) and Dr. Alejandro Benjamin Balazs (Harvard University) and available on BEI resources. The titration of the stock batch (Lot: 70042784) was $4.99 \times 10^5$ relative luciferase tissue culture infectious dose 50% (TCID$_{50}$) units in HEK293-hACE2 cells at 48 hpi.

## Virus infection assays

For authentic SARS-CoV-2, cells were seeded a day prior to infection. The culture supernatant was replaced with serum free DMEM prior to infection. Then, the cells were incubated with SARS-CoV-2 at appropriate MOI for 1 hr. Then, the virus inoculum was aspirated. The cells were washed twice with DPBS (Gibco, Cat. no. 14190144). Then, cells were replenished with serum free DMEM. The culture supernatants or infected cells were harvested at indicated time points.

For ΔS-VRP(G), cells were infected as previously described [41,101] and similarly as that of the authentic SARS-CoV-2 but with 2 hr virus inoculation instead of 1 hr. After virus inoculation, infected cells were washed twice with DPBS. Cells were then replenished with DMEM/F12 with 20% FBS, 100 U/mL penicillin, and 100 µg/mL streptomycin. The culture supernatants or infected cells were harvested at indicated time points.

For HIV-1-spike-Luc/ZsGreen, cells were infected as previously described [43,44]. Cells were incubated with the virus in the presence of 5 µg/ml polybrene for 1 hr in DMEM/F12 with 20% FBS, 100 U/mL penicillin, and 100 µg/mL streptomycin. Then, virus inoculum was removed. Fresh DMEM/F12 with 20% FBS, 100 U/mL penicillin, and 100 µg/mL streptomycin was added. At 48 hpi, infected cells were harvested for downstream analysis.

## Plaque assay

A day before infection, $4 \times 10^5$ or $8 \times 10^5$ Vero-hTMPRSS2 cells were seeded respectively on 12- or 6- well plates. Vero-hTMPRSS2 cells were infected with serially diluted viral stocks for 1 hr. Then, the virus inoculum was removed. Infected cells were overlayed with 1% agarose in DMEM/ DPBS (1:1). At 72 hpi, infected cells were overnight fixed with 4% paraformaldehyde/PBS (ThermoFisher Scientific, Cat. no. J19943.K2). Then, the overlay was removed, and the cells were stained with 0.5% crystal violet in 10% methanol for at least 5 min. Plaques were visualized. Virus concentration was calculated based on the number of plaques in the countable dilution.

## Viral copy number detection

Viral RNA was extracted by RNeasy Plus kit (QIAGEN, Cat. no. 74134) in which viral containing solutions and RLT plus lysis buffer was in 1:4 ratio. Viruses were lysed in RLT plus lysis buffer for 10 min at room temperature. The purified viral RNA was reverse transcribed using the iScript cDNA Synthesis Kit (BIO-RAD, Cat. no. 1708891). Viral RNA was detected by the qPCR primer pair targeting the viral RdRP (S5 Table) with iTaq Universal SYBR Green Supermix (BIO-RAD, Cat. no.1725124) [102,103]. Standard curve for the number of single stranded DNA against Ct value was generated using the BAC-SARS-CoV-2 DNA. The copy number of single stranded RNA was estimated from the copy number of cDNA in each sample.

## Genomic DNA sequencing

The genome DNA of Ctrl, KO1 and KO2 cells were purified with DNeasy Blood and Tissue Kits for DNA Isolation (QIAGEN, Cat. no. 69504). Exon 1 and exon 4 regions encompassing the Cas9 cleavage sites were amplified by Phusion High-Fidelity DNA Polymerase (NEB, Cat. no. M0530L). The sequences of the PCR primer pairs for amplifying exon 1 and exon 4 were listed in S5 Table. Then, the PCR products were submitted for Sanger sequencing services provided by Genomics Division of Iowa Institute of Human Genetics (University of Iowa) using the forward primers.

## Cellular dNTP detection

Cellular dATP, dTTP, dGTP and dCTP levels were measured by a single nucleotide RT incorporation assay previously described [37,40,91]. In brief, $2 \times 10^6$ cells were harvested for dNTP extraction with 65% methanol. Methanol extract was dried at 80°C for 1–2 hr and then reconstituted in reaction buffer at appropriate dilution within linear range of the assay as previously described for dATP, dTTP, dGTP and dCTP levels detection. The levels of cellular dATP, dTTP, dGTP and dCTP were calculated by extrapolating the relative band intensity to the standard curve created by the parallel reactions.

## ActD treatment assay

ActD treatment assay was performed as previously described [57,58]. In brief, Calu-3 Ctrl, KO1 or KO2 cells were treated with 10 µg/mL actD. Cellular RNA was collected at indicated time points. For 0 time point, cells were treated with actD and cellular RNA was extracted immediately after the addition of the drug. Cellular RNA (1 µg) was reverse transcribed and quantified with qPCR by specific primer pairs. The levels of mRNA at indicated time points were normalized to time point 0 hr. One-phase decay curves were plotted by the build-in non-linear regression function of GraphPad Prism.

## Western blot analysis

Upon harvest, cells were washed once with DPBS and lysed with 1 x cell lysis buffer (Cell Signaling Technology, Cat. no. 9803) with phosphatase and protease inhibitor (ThermoFisher Scientific, Cat. no. A32961) for 10–30 min until complete lysis. Cell debris was removed by centrifugation at 16.1 kg for 10 min at 4°C. Total protein concentration was estimated by Pierce BCA Protein Assay Kit (ThermoFisher Scientific, Cat. no. 23225). Protein samples with normalized total protein levels were mixed with 5 x protein sample buffer (0.25 M Tris-Cl pH 6.8, 20% glycerol, 10% SDS and 1% saturated bromophenol blue) to the final concentration of 1 x and heat-denatured at 98°C for 10 min. Cellular protein (10–25 µg) was loaded onto polyacrylamide gels for Western blot analysis as previous described [95]. Primary antibodies used were as follows: Anti-GAPDH (BIORAD, Cat. no. AHP1628), anti-viral N (BEI resources, Cat. no. NR-56223), anti-viral S (ThermoFisher Scientific, Cat. no. PA5114451), anti-SAMHD1 (Abcam, Cat. no. ab117908), anti-STAT1 (Cell Signaling Technology, Cat. no. 9172T), anti-p-STAT1 (Cell Signaling Technology, Cat. no. 8826T), anti-ACE2 (Cell Signaling Technology, Cat. no. 92485T), anti-TMPRSS2 (Santa Cruz Biotechnology, Cat. no. sc-515727), anti-HNF1α (Cell Signaling Technology, Cat. no. 89670T), anti-HNF1β (Proteintech, Cat no. 12533–1-AP) and anti-HNF4α (Cell Signaling Technology, Cat. no.3113T). For secondary antibody, goat anti-mouse IgG (H+L) HRP (Promega, Cat. no. W4021) or goat anti-rabbit IgG (H+L) HRP (Promega, W401B) was used. Immunoblots were developed with Odyssey Fc Imager. Protein band intensity was quantified by Image Studio.

## RNA extraction and RT-qPCR

Cellular RNA extraction was performed with RNeasy Plus Kits according to the manufacturer's instructions. Upon harvest, cells were washed once with DPBS and lysed with RLT plus lysis buffer. After column purification, 1 µg of purified RNA was used for reverse transcription through iScript cDNA Synthesis Kit. The cDNA product was used for qPCR quantification with iTaq Universal SYBR Green supermix. SYBR signal was quantified by QuantStudio 3 Real-Time PCR or CFX Duet Real-Time PCR System. The qPCR primer pairs were either previously designed (including Viral E [104], 18S RNA [105], IL-6 [25], OAS1 [106], ACE2 and dACE2 [55]), predesigned by PrimerBank [107] or designed in-house based on NCBI primer-BLAST with Primer3 adjustment (S5 Table).

## Luciferase detection

To detect secreted *Guassia* luciferease activity, 20 µl of culture supernatant was incubated with 50 µl of the working solution of Pierce Gaussia Luciferase Flash Assay Kit (ThermoFisher Scientific, Cat. no. 16158) for quantification of the

resulting chemiluminescence following manufacturer's instruction. To detect cellular firefly luciferase 2 activity, cells were washed once with DPBS and lysed in 1 x passive lysis buffer (Promega, Cat. no. E1941) supplemented with phosphatase and protease inhibitor. At 30 min lysis, cell debris were removed by centrifugation at 16.1 kg for 10 min at 4°C. Total protein concentration was estimated by Pierce BCA Protein Assay Kit. 10 μl of the PLB lysate was incubated with 100 μl of LAR-II substrate reagent of the Dual-Luciferase Reporter Assay System following manufacturer's instruction to quantify the resulting chemiluminescence (Promega, Cat. no. E1980). The cellular firefly luciferase 2 activities were calculated as follow: ($RLU_{infected}$ per one microgram protein) – ($RLU_{uninfected}$ per one microgram protein).

### dN supplementation assay

Calu-3 cells cultured in complete medium were treated with dN in 1:10 dilution ratio, which consisted of a mixture of 12.5 mM of each dA (Sigma-Aldrich, Cat. no. D8668), dC (Sigma-Aldrich, Cat. no. D0776), dG (Sigma-Aldrich, Cat. no. D0901) and dT (Sigma-Aldrich, Cat. no. T1895) dissolved in blank DMEM. The final concentration for each dN was 1.25 mM. Cells were collected at indicated time points for downstream analysis. For SARS-CoV-2 infection experiment, Calu-3 cells were pretreated with dN for 20 hr. Then, cells were infected with SARS-CoV-2 at MOI = 1 in the absence of dN for 1 hr in serum free DMEM. Next, cells were washed twice with DPBS. 1 mL DMEM with or without dN was replenished to the infected cells. At 24 hpi, culture supernatant and cells were collected for plaque assay analysis and RT-qPCR analysis.

### mRNA-seq

Calu-3 Ctrl, KO1 and KO2 cells ($8 \times 10^5$) were seeded in 6 well plates in biological triplicate. Cells were cultured in 2 mL DMEM/F12 with 20% FBS, 100 U/mL penicillin, and 100 μg/mL streptomycin. At 20 hr after seeding, cells were washed once with DPBS and lysed with RLT plus lysis buffer. Cellular RNA was purified with RNeasy Plus kit following manufacturer's instructions. All RNA samples got RNA integrity Number (RIN) scores equal to or more than 9.9 as determined by Agilent BioAnalyzer by Genomics Division of Iowa Institute of Human Genetics (University of Iowa). The subsequent steps of the mRNA-seq were performed by the Genomics Division using manufacturer recommended protocols. Briefly, at least 500 ng of DNase I-treated total RNA was used to prepare sequencing libraries using the Illumina stranded mRNA library preparation kit (Illumina, Cat. no. 20040534). Barcoded libraries were quantified using a Qubit fluorometer (ThermoFisher Scientific) and pooled together to give the same molar concentration per sample. The pooled libraries were sequenced on the Element AVITI24 sequencing platform on a Cloudbreak FS 150 cycle (2 x 75 bp) medium output sequencing flow cell (Element Biosciences, Cat. no. 860–00014) to generate at least 30M paired-end sequencing reads per sample. The raw base files were converted to FASTQ files using the Bases2fastq v2.1 tool. The mRNA-seq data was deposited in the Gene Expression Omnibus (GEO) with an accession number GSE311600. (released on November 30, 2025): https://www.ncbi.nlm.nih.gov/geo/query/acc.cgi?acc=GSE311600

### Bioinformatic analysis for mRNA-seq

The bioinformatic analysis to the raw FASTQ data obtained from mRNA-seq was performed by Bioinformatics Division, Iowa Institute of Human Genetics (University of Iowa). In brief, PCA including PCA plot, PCA biplot and DEG analysis were generated by SALMON DeSeq2. Pathway analysis including GOEA was performed by iPathwayGuide platform (ADVAITABIO).

### Statistical analysis

At least biological triplicate was performed for each experiment throughout the study. Statistical analysis was performed by GraphPad Prism. For the dataset of the mRNA-seq experiment, FDR-adjusted P value was obtained. Only changes that had P values less than 0.05 were considered statistically significant.

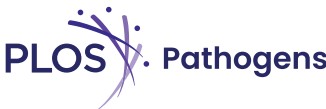

## Supporting information

**S1 Fig. Validation of SAMHD1 deficiency in SAMHD1 KO Calu-3 cells.** (A and B) The Sanger sequencing results to the regions of SpCas9 cleavage sites directed by respectively sgRNA1 or sgRNA2 of (A) KO1 and (B) KO2 cells were shown and compared with that of the Ctrl cells. Red arrows indicated one base pair insertion near the SpCas9 cleavage site. (C) dATP, dTTP, dCTP and dGTP levels of Calu-3 Ctrl, KO1 or KO2 cells were measured with HIV-1 RT based single nucleotide incorporation assay. For (C), one-way ANOVA multiple comparisons test was used to evaluate the statistical significance of the difference between Ctrl and KO1 or KO2. ** $P < 0.01$, *** $P < 0.001$.
(TIF)

**S2 Fig. Independent experiments for quantification of viral N, p-STAT1 and STAT1 proteins of SARS-CoV-2-infected SAMHD1 KO Calu-3 cells upon baricitinib treatment.** (A) The experiment described in Fig 2F was repeated independently three times. (B) The relative band intensities of viral N, STAT1 or p-STAT1 as shown in S2A Fig were calculated by dividing them with GAPDH and were normalized to Ctrl cells treated with DMSO (-). One-way ANOVA multiple comparisons test was used to evaluate the statistical significance of the difference upon baricitinib treatment. * $P < 0.05$, ** $P < 0.01$. ns, not significant.
(TIF)

**S3 Fig. Independent experiments for quantification of the viral N protein of ΔS-VRP(G)-infected SAMHD1 KO Calu-3 cells.** (A) Calu-3 Ctrl, KO1 or KO2 cells were infected with ΔS-VRP(G) at $1.7 \times 10^4$ copy number per cell. Viral N protein was detected with Western blot at indicated time points. Arrows indicate the protein band of viral N protein. GAPDH was used as input control. Three independent experiments were performed. (B) The relative band intensities of viral N protein as shown in S3A Fig were calculated by dividing them with GAPDH bands and normalized to Ctrl cells at 24 hpi. Two-way ANOVA multiple comparisons test was used to evaluate the statistical significance of the difference between Ctrl and KO1 or KO2. ns, not significant.
(TIF)

**S4 Fig. PCA clustering of Calu-3 Ctrl, KO1 and KO2 cells based on the mRNA-seq profiles.** Nine samples of the mRNA-seq analysis gave nine PCs (i.e., Ctrl, KO1 and KO2 cells, biological triplicate). The influence of each PC was plotted based on their contributing variance (%). The cumulative variances from 1 to 9 were plotted above and joined by a curve. PC1 and PC2 gave 98% coverage of all PCs. (B) Individual samples from Ctrl, KO1 and KO2 cells were plotted along the PC1 and PC2 based on the net scores of loading genes obtained from PCA. (C) A PCA biplot was generated by top ten loading genes influencing PC1 and PC2.
(TIF)

**S5 Fig. Independent experiments for quantification of ACE2, TMPRSS2, HNF1α, HNF1β and HNF4α proteins of SAMHD1 KO Calu-3 cells.** (A) ACE2, TMPRSS2, HNF1α, HNF1β and HNF4α proteins of Ctrl, KO1 and KO2 cells were detected by Western blot. Triangle (◁) and asterisk (*) symbols respectively represented the zymogen (~55 kDa) and active form (~31 kDa) of TMPRSS2. Three independent experiments were performed. GAPDH was detected as input control. The relative band intensities of (B) ACE2, (C) TMPRSS2 (◁+*), (D) HNF1α, (E) HNF1β and (F) HNF4α as shown in S5A Fig were calculated by dividing them with GAPDH bands and were normalized to Ctrl cells. One-way ANOVA multiple comparisons test was used to evaluate the statistical significance of the difference between Ctrl and KO1 or KO2. * $P < 0.05$, ** $P < 0.01$, *** $P < 0.001$, **** $P < 0.0001$. ns, not significant.
(TIF)

**S6 Fig. Confirming *ACE2* and *dACE2* mRNA expressions in SAMHD1 KO Calu-3 cells.** The mRNA levels of (A) *ACE2* and (B) *dACE2* of Ctrl, KO1 and KO2 cells were measured by RT-qPCR with qPCR primer pairs previously described [55].

For ACE2, primer pair 4 listed in S5 Table was used. Biological triplicate experiment was performed. One-way ANOVA multiple comparisons test was used to evaluate the statistical significance of the difference between Ctrl and KO1 or KO2. ** P<0.01, *** P<0.001. ns, not significant.
(TIF)

**S7 Fig. Independent experiments performed for protein band calculation of Figs 7B and 8B-8D.** The Western blot results to two more independent experiment replicates for the same experiment conducted in Figs 7A and 8A were respectively shown in (A) and (B).
(TIF)

**S8 Fig. Gene ontology enrichment analysis to cellular components to the common DEGs of SAMHD1 KO Calu-3 cells.** 681 consistent DEGs obtained from SAMHD1 KO Calu-3 cells as shown in Fig 4A were used for gene ontology enrichment analysis to the category of "Cellular component". Top ten GO terms with the best FDR-adjusted P values were plotted on a bubble plot against the rich ratio (i.e. number of DEGs per all annotated genes of the GO term). The number of counts of DEGs was shown in terms of the size of the bubble. The bubble plot was generated with the online resources SR plot (https://www.bioinformatics.com.cn/srplot).
(TIF)

**S1 Table. DEGs observed in the mRNA-seq data sets on the three comparison pairs (Ctrl vs KO1, Ctrl vs. KO2 and KO1 vs. KO2 cells).**
(XLSX)

**S2 Table. The 681 common DEGs observed in both Calu-3 KO1 and KO2 cells as shown in Fig 4A.**
(XLSX)

**S3 Table. All GO terms in the category of "Cellular component" enriched based on the 681 common DEGs shown in Fig 4A.**
(XLSX)

**S4 Table. Enriched DEGs of the top ten GO terms as shown in S8 Fig.**
(XLSX)

**S5 Table. Templates and oligonucleotide sequences and related information.**
(XLSX)

## Acknowledgments

We thank Drs. Balaji Manicassamy and Patrick Sinn for their invaluable support during this study. We thank Drs. Jesse Bloom and Alejandro Benjamin Balazs for the resources donated to Biodefense and Emerging Infections Research Resources Repository (BEI Resources). The following reagents were obtained through BEI Resources, NIAID, NIH: (1) SARS-related coronavirus 2, Wuhan-Hu-1 spike-pseudotyped lentivirus, Luc2/ZsGreen, NR-53818, and (2) Monoclonal anti-SARS-related coronavirus 2 nucleocapsid protein (produced in vitro), NR-56223. The mRNA seq data herein were obtained at the Genomics Division (RRID: SCR_023422) and the Bioinformatics Division of the Iowa Institute of Human Genetics which are supported, in part, by the University of Iowa Carver College of Medicine. We also thank Mr. Fernando William Moreira Santana for technical assistance. We appreciate valuable discussions with the members of Dr. Wu's, Dr. Perlman's, and Dr. Sinn's laboratories at the University of Iowa, the members of Dr. Kim's laboratory at Emory University as well as Dr. Constanza Espada from the University of Missouri.

## Author contributions

**Conceptualization:** Pak-Hin Hinson Cheung, Li Wu.

**Data curation:** Pak-Hin Hinson Cheung, Krisztina Ambrus, Shravya Honne.

**Formal analysis:** Pak-Hin Hinson Cheung, Krisztina Ambrus, Shravya Honne.

**Funding acquisition:** Baek Kim, Stanley Perlman, Li Wu.

**Investigation:** Pak-Hin Hinson Cheung, Krisztina Ambrus, Shravya Honne.

**Methodology:** Pak-Hin Hinson Cheung, Pearl Chan, Hua Yang, Baek Kim, Li Wu.

**Project administration:** Baek Kim, Li Wu.

**Resources:** Baek Kim, Stanley Perlman, Li Wu.

**Supervision:** Baek Kim, Stanley Perlman, Li Wu.

**Visualization:** Pak-Hin Hinson Cheung, Hua Yang, Li Wu.

**Writing – original draft:** Pak-Hin Hinson Cheung, Li Wu.

**Writing – review & editing:** Pak-Hin Hinson Cheung, Pearl Chan, Hua Yang, Stanley Perlman, Li Wu.

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
