## [Decision Letter · Decision Letter 0]

17 Dec 2025

SAMHD1 promotes SARS-CoV-2 infection by enhancing HNF1-dependent ACE2 expression in lung epithelial cells

PLOS Pathogens

Dear Dr. Wu,

Thank you for submitting your manuscript to PLOS Pathogens. After careful consideration, we feel that it has merit but does not fully meet PLOS Pathogens's publication criteria as it currently stands. Therefore, we invite you to submit a revised version of the manuscript that addresses the points raised during the review process.

We look forward to receiving your revised manuscript.

Kind regards,

Yong-Hui Zheng, Ph.D.

Guest Editor

PLOS Pathogens

Alexander Gorbalenya

Section Editor

PLOS Pathogens

Editor-in-Chief

PLOS Pathogens

orcid.org/0000-0003-2946-9497

Michael Malim

PLOS Pathogens

orcid.org/0000-0002-7699-2064

**Additional Editor Comments :**

Below comments from Reviewer #1 are very helpful and constructive, they should be addressed by providing new results:

1. In Figure 3B, E RNA is increased in the KO1 and KO2 cells at 24h. This is not addressed in the text. As shown in Figure S1, KO1 and KO2 have elevated dATP levels (and possibly other dNTPs). This could be due to increased dNTP levels in the cells. What happens to SARS-CoV-2 infection and ACE2/TMPRSS2 expression if dNTP levels are artificially increased independent of SAMHD1 expression (i.e., without SAMHD1 knockout)?

2. Given that SARS-CoV-2 infects other epithelial cell lines, it would be worth validating some of these results (SARS-CoV-2 replication and ACE2 protein expression) in at least one other cell line or in primary epithelial cells to ensure that this is not a Calu-3 specific effect.

3. For western blots (Figures 2F, 3C, 4D, 6E), only one experiment is shown. For rigor and reproducibility and because conclusions are based on these results, these experiments should be repeated and the results graphed with error bars and statistics similar to what was done for Figures 7 and 8.

**Journal Requirements:**

At this stage, the following Authors/Authors require contributions: Pak-Hin Hinson Cheung, Pearl Chan, Hua Yang, Shravya Honne, Baek Kim, Stanley Perlman, and Li Wu. Please ensure that the full contributions of each author are acknowledged in the "Add/Edit/Remove Authors" section of our submission form.

- ® on pages: 29, and 33

- TM on pages: 29, 31, 32, and 52.

4) We notice that your supplementary figures are uploaded with the file type 'Figure'. Please upload them separately with the file type  'Supporting Information'. Please ensure that each Supporting Information file has a legend listed in the manuscript after the references list.

Potential Copyright Issues:

i) Figure 9. Please confirm whether you drew the images / clip-art within the figure panels by hand. If you did not draw the images, please provide (a) a link to the source of the images or icons and their license / terms of use; or (b) written permission from the copyright holder to publish the images or icons under our CC BY 4.0 license. Alternatively, you may replace the images with open source alternatives. See these open source resources you may use to replace images / clip-art:

6) Please update your Data Availability Statement in the online submission form to include the accession number of the dataset OR a direct link to access the dataset.

7)  Please ensure that the funders and grant numbers match between the Financial Disclosure field and the Funding Information tab in your submission form. Note that the funders must be provided in the same order in both places as well. Currently, the order of the grants is not exactly the same in both places.

8) Please revise your current Competing Interest statement to the standard "The authors have declared that no competing interests exist.".

**Reviewers' Comments:**

Reviewer's Responses to Questions

**Part I - Summary**

Reviewer #1: Cheung et al. demonstrate that SAMHD1 mediates ACE2 protein expression in Calu-3 cells via transcription factors HNF1a and HNF1b. If the SAMHD1 gene is knocked out in Calu-3 cells or if protein levels of HNF1s are depleted, SARS-CoV-2 infection is significantly reduced. This function is independent of IFN signaling/production. The data reflect the conclusions and were overall compelling and novel. However, SAMHD1 KO leads to increased dATP levels in Calu-3 cells and this phenotype was not characterized for its potential role in ACE2 expression or SARS-CoV-2 infection. Investigation of dNTP levels would strengthen the study.

Reviewer #2: In the manuscript entitled "SAMHD1 Promotes SARS-CoV-2 Infection by Enhancing HNF1-Dependent ACE2 Expression in Lung Epithelial Cells, the authors generated two SAMHD1 knockout (KO) Calu-3 cell clones (KO1 and KO2) and demonstrated that SAMHD1 acts as a proviral factor for SARS-CoV-2 infection. First, they showed that SARS-CoV-2 replication was suppressed in SAMHD1 KO Calu-3 cells, independent of IFN signaling. Second, they confirmed that SAMHD1 supports spike (S) protein-mediated viral entry by maintaining ACE2 expression. Finally, they discovered that HNF1-dependent ACE2 expression was suppressed in SAMHD1 KO Calu-3 cells. In summary, this study reveals a novel function and mechanism of SAMHD1 in supporting SARS-CoV-2 infection in Calu-3 cells. The manuscript is well-structured and clearly presented. The reviewer has several comments.

**Part II – Major Issues: Key Experiments Required for Acceptance**

Reviewer #1: 1. In Figure 3B, E RNA is increased in the KO1 and KO2 cells at 24h. This is not addressed in the text. As shown in Figure S1, KO1 and KO2 have elevated dATP levels (and possibly other dNTPs). This could be due to increased dNTP levels in the cells. What happens to SARS-CoV-2 infection and ACE2/TMPRSS2 expression if dNTP levels are artificially increased independent of SAMHD1 expression (i.e., without SAMHD1 knockout)?

2. Given that SARS-CoV-2 infects other epithelial cell lines, it would be worth validating some of these results (SARS-CoV-2 replication and ACE2 protein expression) in at least one other cell line or in primary epithelial cells to ensure that this is not a Calu-3 specific effect.

3. For western blots (Figures 2F, 3C, 4D, 6E), only one experiment is shown. For rigor and reproducibility and because conclusions are based on these results, these experiments should be repeated and the results graphed with error bars and statistics similar to what was done for Figures 7 and 8.

Reviewer #2: There are three moderate issues noted during the review of this manuscript. The authors are welcome to address them by experiments or detailed discussion.

1. Based on the results of this study, the authors suggested that SAMHD1 expression promotes S protein-mediated viral entry, thereby facilitating infection by authentic SARS-CoV-2. To confirm this finding, the authors should test whether overexpression of SAMHD1 in Ctrl cells further enhances SARS-CoV-2 infection, and whether complementation of SAMHD1 in KO1 or KO2 cells restores SARS-CoV-2 infection.

2. In Figure 5, the reviewer observed that the result curves deviated from specific numerical points. How did the authors generate these figure panels? Please provide more details and make the necessary changes accordingly.

3. Similar to the first major comment, the authors could assess whether overexpression of HNF1α or HNF1β in Calu-3 cells further enhances ACE2 levels and SARS-CoV-2 infection. Additionally, complementation of HNF1α or HNF1β in KO1 or KO2 cells could restore ACE2 levels and SARS-CoV-2 infection.

**Part III – Minor Issues: Editorial and Data Presentation Modifications**

Reviewer #1: 1. Line 20: should be “broad” instead of “board”

2. It is unclear what sgRNA control was used for the SAMHD1 KO in Calu-3 cells. Is this a sgRNA that recognizes a nonhuman gene? Or no sgRNA?

3. Lines 114, 529, 1177: Sanger should be capitalized.

4. Line 1063 and Fig 2B: confirm that “copy number” refers to viral RNA copy number.

5. Lines 192-193: as written, this sentence suggests that KO1 and KO2 cells are clustered together, which is not the case and this should be rewritten.

6. Line 194: it is unclear what is meant by “which however minimally distinguished”.

7. Verb tenses (present vs. past tense) fluctuate throughout the manuscript, which makes it difficult to read.

Reviewer #2: 1. In this study, the authors confirmed that SAMHD1 supports SARS-CoV-2 infection by maintaining HNF-1-dependent ACE2 expression in Calu-3 cells. The authors may consider using the current title, “SAMHD1 promotes SARS-CoV-2 infection by enhancing HNF1-dependent ACE2 expression in lung epithelial cells,” to reflect the results from the overexpression and complementation experiments.

2. Line 105, what is the “HD” domain? Please provide the full name of “HD” when it first appears in the main text.

3. Line 239, the subtitle “SAMHD1 expression does not affect ACE2 mRNA stability” should be changed to “SAMHD1 KO does not affect ACE2 mRNA stability.”

4. Line 274, please provide the full name of “GATA6” when it first shows in the main text. Similarly, there were other unexplained abbreviations that the reviewer did not list. Please make changes carefully.

5. Lines 23-24: The description “Some viruses escape SAMHD1 restriction by utilizing SAMHD1-mediated innate immune suppression to establish effective infection through viral antagonism” is confused. Please revise it for clarity.

PLOS authors have the option to publish the peer review history of their article (what does this mean? ). If published, this will include your full peer review and any attached files.

**Do you want your identity to be public for this peer review?** For information about this choice, including consent withdrawal, please see our Privacy Policy .

Reviewer #1: No

Reviewer #2: No

**Figure resubmission:**

**Reproducibility:**



---

## [Decision Letter · Decision Letter 1]

8 Mar 2026

Dear Prof. Wu,

We are pleased to inform you that your manuscript 'SAMHD1 depletion restricts SARS-CoV-2 infection by suppressing HNF1-dependent ACE2 expression in lung epithelial cells' has been provisionally accepted for publication in PLOS Pathogens.

Best regards,

Yong-Hui Zheng, Ph.D.

Guest Editor

PLOS Pathogens

Alexander Gorbalenya

Section Editor

PLOS Pathogens

Sumita Bhaduri-McIntosh

Editor-in-Chief

PLOS Pathogens

orcid.org/0000-0003-2946-9497

Michael Malim

Editor-in-Chief

PLOS Pathogens

orcid.org/0000-0002-7699-2064

Reviewer Comments (if any, and for reference):

Reviewer's Responses to Questions

**Part I - Summary**

Reviewer #1: Cheung et al. demonstrate that SAMHD1 mediates ACE2 protein expression in Calu-3 cells via transcription factors HNF1a and HNF1b. In Calu-3 cells, SAMHD1 gene knock out, higher dNTP concentrations and HNF1a/b protein depletion lead to significantly reduced SARS-CoV-2 infection. This function is independent of IFN signaling/production. The data reflect the conclusions and were overall compelling and novel.

Reviewer #2: The authors have responded my questions well and all major issues from this reviewer were addressed well.

**Part II – Major Issues: Key Experiments Required for Acceptance**

Reviewer #1: N/A

Reviewer #2: None

**Part III – Minor Issues: Editorial and Data Presentation Modifications**

Reviewer #1: N/A

Reviewer #2: None

PLOS authors have the option to publish the peer review history of their article (what does this mean? ). If published, this will include your full peer review and any attached files.

**Do you want your identity to be public for this peer review?** For information about this choice, including consent withdrawal, please see our Privacy Policy .

Reviewer #1: No

Reviewer #2: No

---

## [Editor Report · Acceptance letter]

Dear Prof. Wu,

We are delighted to inform you that your manuscript, "SAMHD1 depletion restricts SARS-CoV-2 infection by suppressing HNF1-dependent ACE2 expression in lung epithelial cells," has been formally accepted for publication in PLOS Pathogens.

Best regards,

Sumita Bhaduri-McIntosh

Editor-in-Chief

PLOS Pathogens

orcid.org/0000-0003-2946-9497

Michael Malim

Editor-in-Chief

PLOS Pathogens

orcid.org/0000-0002-7699-2064